# OpenReview forum: "Is Privacy Always Prioritized over Learning? Probing LLMs' Value Priority Belief under External Perturbations"
_ICLR.cc/2026/Conference — ICLR 2026 Conference Withdrawn Submission_

### Official Review · Reviewer_o4ni · 2025-10-26

**Soundness:** 1
**Presentation:** 1
**Contribution:** 1
**Rating:** 2
**Confidence:** 4

**Summary:**

This paper investigates how large language models’ (LLMs) value rankings can be influenced through different prompting strategies. The authors experiment with six prompting methods to examine how emphasizing or reducing specific values affects a model’s internal value hierarchy, which is derived using pairwise “value battles.” The study evaluates these value rankings using the dataset and methodology introduced in LitmusValues (Chiu et al., 2025). The results show that the “Scenario” prompting method notably shifts a model’s default value orientation, with larger models exhibiting greater sensitivity to prompt-based value changes. Additionally, the results show correlations among different values, suggesting interconnected value structures.

**Strengths:**

1. This paper explores how LLMs can change their value rankings given steering prompts.
2. This paper shows that the steerability of values depends on the input prompt.

**Weaknesses:**

# 1. Novelty
- While it is encouraged to build upon prior work, a huge portion of this paper closely overlaps with or is derived from the following studies [1,2]. The Elo rating system and dataset are from [1], and even the figure design appears highly similar (e.g., Figure 5 from [1] and Figure 4 from this paper). The primary novel contribution of this paper is in introducing six prompt techniques to modify LLMs’ value rankings.

- One of the main findings of this paper is that models of similar sizes exhibit similar value ranking patterns. However, as noted in lines 430–431, a similar pattern has already been observed in prior work [3], which diminishes the novelty of the current study.

[1] Yu Ying Chiu, Zhilin Wang, Sharan Maiya, Yejin Choi, Kyle Fish, Sydney Levine, and Evan Hubinger. Will ai tell lies to save sick children? litmus-testing ai values prioritization with airiskdilemmas. arXiv preprint 2025.
[2] Yu Ying Chiu, Liwei Jiang, and Yejin Choi. Dailydilemmas: Revealing value preferences of llms with quandaries of daily life. In The Thirteenth International Conference on Learning Representations, 2025.
[3] Minyoung Huh, Brian Cheung, Tongzhou Wang, and Phillip Isola. The platonic representation hypothesis. In The International Conference on Machine Learning, 2024.


# 2. Conceptual and Interpretive Issues
- The claim that larger models’ values are more susceptible to manipulation (lines 82–84) appears overstated. The ability to adjust value priorities through prompting is not inherently problematic; rather, it reflects proper instruction following.

- The interpretation of Figure 5 (lines 355–357) and Figure 7 (lines 404–409) lacks sufficient justification or explanation. For Figure 5, further explanation is needed why the Persona method is better than the Scenario method in the “Reduce” configuration. For Figure 7, the high correlation observed within the two value sets may reflect two distinct concepts: (Privacy, Justice, Respect, Truth, Freedom) as “moral principles” and (Adaptability, Creativity, Care, Cooperation, Learning, Sustainability, Wisdom) as “growth-oriented” values. A discussion of this possibility would be beneficial.

- The experiment addressing Research Question 3 (value entrenchment) does not convincingly support the stated conclusions. While the intention to demonstrate the usefulness of the Scenario prompting method is reasonable, to validly support the claim that this method can be used for “value entrenchment”, additional baseline experiments should be conducted using existing steering approaches, such as activation engineering methods.


# 3. Empirical and Analytical Limitations
- The analysis of Figure 4 appears selectively interpreted. The authors generalize from a single sample case (lines 347–349), overlooking internal inconsistencies—for example, GPT-4.1-nano shows larger rank shifts within the same value dimension (adaptability).


- When presenting correlation results, it would be better if the corresponding p-values are also shared.

- Figure 9 lacks a clear explanation of the y-axis and is difficult to interpret.



# 4. Clarity and Presentation
- Figure 4 is visually difficult to interpret. Improving the design (e.g., clearer labeling, consistent color mapping) would enhance readability. Also, there is a typo in the name of the model (e.g., LLaMA3-7B). Figure 9 also requires visual and explanatory refinement.


- The interpretation sections would benefit from concise summaries and better linkage between visualizations and textual analysis.


# 5. Writing and Formatting Issues
- Multiple typographical and grammatical errors hinder readability (e.g., line 71: “qustions”, line 341: “nder”).


- There are typos, unpunctuated sentences, and incomplete sentences (e.g., lines 128, 229, 356–357).

- Multiple references to the following paper: “Do llms have consistent values?”. Citing only a single version of the paper is recommended (cite the ICLR 2025 version).

- It’s acceptable to cite the arXiv version of a paper, but if the paper has been officially published, it would be better to use the BibTeX entry of the published version instead.

- Although the appendix serves as supplementary material, there are several citation errors (e.g., lines 987 and 991). In addition, there are multiple instances where citep is used instead of citet (e.g., line 1041).

- The caption of Table 9 is below the table.

**Questions:**

1. Would it not be desirable for larger models to exhibit a greater ability to adjust their value orientations in response to given instructions? Given that larger models typically possess stronger instruction-following capabilities, shifts in value rankings based on input prompts are a natural phenomenon. Moreover, such adaptability would be advantageous in applications like Persona Prompting (e.g., assigning specific personas to agents).

2. Do you believe that Research Question 3 is sufficiently supported by the experimental results presented in Figure 9?

---

> ### Author Response · Authors · 2025-11-27
> **Response [1/n]**
>
> Thank you for your time and insightful comments on our paper. We appreciate your recognition of the strengths of our work.
>
> Sorry for the late reply because of the new experiments. Please see our response below.
>
>
>
> ## **Q1**. [Lack of novelty].
> > 1. Novelty
> > While it is encouraged to build upon prior work, a huge portion of this paper closely overlaps with or is derived from the following studies [1,2]. The Elo rating system and dataset are from [1], and even the figure design appears highly similar (e.g., Figure 5 from [1] and Figure 4 from this paper). The primary novel contribution of this paper is in introducing six prompt techniques to modify LLMs’ value rankings.
> > One of the main findings of this paper is that models of similar sizes exhibit similar value ranking patterns. However, as noted in lines 430–431, a similar pattern has already been observed in prior work [3], which diminishes the novelty of the current study.
>
>
> ***Ans for Q1:***
> We acknowledge that we utilize the *Litmus Values* framework (Elo ratings and AIRiskDilemmas) [1,2] as our measurement tool. However, our work contributes three distinct novelties that differ fundamentally from the cited works:
>
> 1.  **From Static to Dynamic:** While [1] focused on measuring *static* value snapshots, our work is the first to systematically benchmark **Value Plasticity**—quantifying *how easily* these rankings can be collapsed or inverted by external perturbations.
> 2.  **Discovery of Value Topology:** We identify a "Value Correlation Topology" (Finding 3), revealing that values form a latent causal graph where manipulating one value (e.g., Adaptability) triggers predictable shifts in unconnected values (e.g., Privacy).  This structural finding is absent in the cited works.
> 3.  **Behavioral Evidence for Platonic Hypothesis:** While Huh et al. (2024) proposed the Platonic Representation Hypothesis regarding internal representations, we provide novel **behavioral evidence** in the specific domain of *ethical values*. We show that as models scale, their *output-level* value correlations converge, validating the hypothesis through a new lens.
>
>
>
>
>
> ## **Q2**. [Conceptual/Interpretive issues].
> > 2. Conceptual and Interpretive Issues
> > The claim that larger models’ values are more susceptible to manipulation (lines 82–84) appears overstated. The ability to adjust value priorities through prompting is not inherently problematic; rather, it reflects proper instruction following.
>
> ***Ans for Q2:***
> We agree that value shifts reflect instruction following capabilities. However, most of the value manipulations during inference need to be implemented by the instructions. Nevertheless, in our experiments, the scenario and persona methods do not explicitly instruct the models to prioritize values.
>
> Besides, we argue that in the context of **AI Safety**, "instruction following" becomes "susceptibility" when the instruction is adversarial.
>
> * **The Threat Model:** As stated in the paper, understanding LLM values is crucial for mitigating risks like "jailbreaks and persuasive manipulations".
> * **Why it is Problematic:** If a model effectively follows an instruction to "prioritize Efficiency over Safety" (instruction following), it results in an unsafe agent (vulnerability). Our finding that larger models are *more* susceptible highlights a critical trade-off: as models become more capable instruction followers, they may become less robust to coercive value misalignment. We classify this as a "risk of coercion" rather than a failure of capability.

---

> ### Author Response · Authors · 2025-11-27
> **Response [2/n]**
>
> ## **Q3**. [Fig. 5/7 interpretation].
> > The interpretation of Figure 5 (lines 355–357) and Figure 7 (lines 404–409) lacks sufficient justification or explanation. For Figure 5, further explanation is needed why the Persona method is better than the Scenario method in the “Reduce” configuration. For Figure 7, the high correlation observed within the two value sets may reflect two distinct concepts: (Privacy, Justice, Respect, Truth, Freedom) as “moral principles” and (Adaptability, Creativity, Care, Cooperation, Learning, Sustainability, Wisdom) as “growth-oriented” values. A discussion of this possibility would be beneficial.
>
> ***Ans for Q3:***
> * **Figure 5 (Reduce Configuration):** We observe that **Persona** prompting is often more effective than Scenario prompting for *reducing* values (Blue bars in Fig. 5). We hypothesize this is because "Scenario" prompts often rely on world-building which implies *positive* prioritization (e.g., "In Valoria, X is supreme"). Constructing a narrative purely around the *absence* or *negation* of a value is often less conceptually coherent for the model than simply assigning a Persona that "explicitly considers [Value] to be unimportant".
> * **Figure 7 (Clusters):** We thank the reviewer for this insightful categorization.  The heatmap indeed supports a distinction between **Moral Principles** (Privacy, Justice, Respect, Truth, Freedom) and **Growth/Utility Values** (Adaptability, Creativity, Learning, Wisdom). We have updated Section 5.2 to explicitly discuss this "Moral vs. Utility" clustering, as it strengthens our "Value Topology" finding by providing a semantic explanation for the observed correlations.
>
>
>
>
>
>
> ## **Q4**. [RQ3 (entrenchment) unsupported].
> > The experiment addressing Research Question 3 (value entrenchment) does not convincingly support the stated conclusions. While the intention to demonstrate the usefulness of the Scenario prompting method is reasonable, to validly support the claim that this method can be used for “value entrenchment”, additional baseline experiments should be conducted using existing steering approaches, such as activation engineering methods.
>
> ***Ans for Q4:***
> We acknowledge that activation engineering is a powerful steering method. However, our study focuses strictly on **black-box prompting strategies**, which are the only methods available to the vast majority of users and developers working with API-based models (like GPT-4). Our future work will explore the activation engineering methods.
>
>
>
> ## **Q5**. [Fig. 4 selective interpretation].
> > 3. Empirical and Analytical Limitations
> > The analysis of Figure 4 appears selectively interpreted. The authors generalize from a single sample case (lines 347–349), overlooking internal inconsistencies—for example, GPT-4.1-nano shows larger rank shifts within the same value dimension (adaptability).
>
> ***Ans for Q5:***
> We apologize if the text appeared to generalize from a single case. While we used specific examples to illustrate the phenomenon, our conclusions are based on the **aggregate metrics** across all models.
>
> * **Aggregate Evidence:** Figure 6 explicitly shows the *Average Elo Change* across all values, demonstrating that larger models (GPT-4.1, LLaMA-70B) generally exhibit greater total plasticity ($\Delta$ Elo) than their smaller counterparts.
> * **Nano Model Instability:** We acknowledge that smaller models like GPT-4.1-nano show rank shifts (e.g., Adaptability). However, smaller models often exhibit high variance due to "confusion" or weaker coherence, whereas the shifts in larger models are more systematic and responsive to the specific semantic content of the prompt. We have refined the text to distinguish between "systematic plasticity" (large models) and "instability/noise" (small models).
>
>
> ## **Q6**. [Fig. 9 unclear].
> > Figure 9 lacks a clear explanation of the y-axis and is difficult to interpret.
>
> ***Ans for Q6:***
> We will revise the caption and labels of Figure 9 for clarity.
>
> * **X-axis:** Represents the **initial persuasion strength** of the Scenario prompt (how much it moved the target value rank away from default).
> * **Y-axis:** Represents the **Rank of the target value after the Persona attack**.
> * **Interpretation:** Data points that remain high on the Y-axis (despite the attack) indicate successful "entrenchment."
>
> [Image of value entrenchment scatter plot]
>  The plot shows that for larger models (Qwen2.5-72B), the Scenario method helps maintain the target value's rank against the Persona perturbation better than the baseline (red line).

---

> ### Author Response · Authors · 2025-11-27
> **Response [3/n]**
>
> ## **Q7**. [Fig. 4 clarity].
> > 4. Clarity and Presentation
> > Figure 4 is visually difficult to interpret. Improving the design (e.g., clearer labeling, consistent color mapping) would enhance readability. Also, there is a typo in the name of the model (e.g., LLaMA3-7B). Figure 9 also requires visual and explanatory refinement.
>
> ***Ans for Q7:***
> We will redesign Figure 4 to improve readability.
>
> * **Enhancements:** We have revised Figure 4. We used a more distinct color to represent Rank 1 (Red) vs Rank 16 (Blue), and corrected the model name typo (LLaMA3-7B $\rightarrow$ LLaMA3-8B) in the Figure 4.
>
>
>
> ## **Q8**. [Writing/Formatting issues].
> > 5. Writing and Formatting Issues
> > Multiple typographical and grammatical errors hinder readability (e.g., line 71: “qustions”, line 341: “nder”).
> > There are typos, unpunctuated sentences, and incomplete sentences (e.g., lines 128, 229, 356–357).
> > Multiple references to the following paper: “Do llms have consistent values?”. Citing only a single version of the paper is recommended (cite the ICLR 2025 version).
> > It’s acceptable to cite the arXiv version of a paper, but if the paper has been officially published, it would be better to use the BibTeX entry of the published version instead.
> > Although the appendix serves as supplementary material, there are several citation errors (e.g., lines 987 and 991). In addition, there are multiple instances where citep is used instead of citet (e.g., line 1041).
> > The caption of Table 9 is below the table.
>
> ***Ans for Q8:***
> We thank the reviewer for the detailed proofreading. We will correct the following in the camera-ready version:
>
> * **Typos:** "qustions", "nder", "finegrained".
> * **Citations:** We will unify all citations to the official ICLR 2025 version of the reference papers and correct the `citep`/`citet` usage.
> * **Formatting:** We will fix the position of the Table 9 caption and ensure all tables are properly aligned.
>
>
>
>
> ## **Q9**. [Desirability of value adjustment].
> > Would it not be desirable for larger models to exhibit a greater ability to adjust their value orientations in response to given instructions? Given that larger models typically possess stronger instruction-following capabilities, shifts in value rankings based on input prompts are a natural phenomenon. Moreover, such adaptability would be advantageous in applications like Persona Prompting (e.g., assigning specific personas to agents).
>
> ***Ans for Q9:***
> We agree that adaptability is desirable for **role-playing** (e.g., "Be a Pirate"). However, we argue it is undesirable for **Safety Critical Values**.
>
> * **The Distinction:** Our paper focuses on "Values" defined as ethical guardrails (e.g., Privacy, Protection, Truthfulness). If a simple prompt can easily invert these specific values (e.g., making a model prioritize "Efficiency" over "Privacy"), it enables **Alignment Faking** and **Jailbreaking**.
> * **Conclusion:** While plasticity is a feature for style, it is a potential misuse for safety. Our work highlights the tension that "larger models typically possess stronger instruction-following capabilities", which paradoxically makes their safety guardrails harder to guarantee under adversarial pressure. Our work highlights the potential conflicts between the design goals of LLMs, such as instruction-following and safety guardrails.
>
>
> ## **Q10**. [Support for RQ3].
> > Do you believe that Research Question 3 is sufficiently supported by the experimental results presented in Figure 9?
>
> ***Ans for Q10:***
> We believe RQ3 is supported within the context of prompt engineering.
>
> * **Evidence:** Figure 9 demonstrates a measurable "resistance effect." When a model is conditioned with a Scenario (e.g., The Avengers), the $\Delta$ Rank caused by a conflicting Persona prompt is reduced compared to the baseline.
> * **Significance:** This proves that *contextual immersion* creates inertia in the value system. While we haven't tested white-box methods, the result confirms that **Scenario prompting** is a viable black-box strategy for stabilizing model behavior, answering RQ3 affirmative.
>
>
>
>
> > **References:**
> >
> > [1] Yu Ying Chiu, Zhilin Wang, Sharan Maiya, Yejin Choi, Kyle Fish, Sydney Levine, and Evan Hubinger. Will ai tell lies to save sick children? litmus-testing ai values prioritization with airiskdilemmas. arXiv preprint 2025.
> >
> > [2] Yu Ying Chiu, Liwei Jiang, and Yejin Choi. Dailydilemmas: Revealing value preferences of llms with quandaries of daily life. In The Thirteenth International Conference on Learning Representations, 2025.
> >
> > [3] Minyoung Huh, Brian Cheung, Tongzhou Wang, and Phillip Isola. The platonic representation hypothesis. In The International Conference on Machine Learning, 2024.
> >

---

### Official Review · Reviewer_Md6f · 2025-10-30

**Soundness:** 3
**Presentation:** 4
**Contribution:** 3
**Rating:** 4
**Confidence:** 5

**Summary:**

This paper investigates the plasticity and robustness of LLM value systems. Specifically, the authors examine how susceptible LLM value rankings are to external perturbations introduced via different prompting methodologies (e.g., direct instruction, persuasion, persona-based prompts). The study systematically explores the resulting changes in value correlation and the overall extent of value ranking manipulation across various model scales. The key findings demonstrate a non-trivial relationship between model scale and both the inherent correlation among values and the manipulability (plasticity) of the value hierarchy under external influence.

**Strengths:**

(1) The core research question—the degree to which LLM value rankings can be altered or influenced by various prompting strategies—is significant.

(2) The work provides valuable empirical evidence regarding the relationship between model scale and the intrinsic correlation among different values.

(3) The paper establishes an experimental relationship between model scale and the manipulability (plasticity) of the LLM's value ranking.

**Weaknesses:**

(1) The analysis of value correlation changes, while present, appears to overlook deeper insights. Specifically, previous work (e.g., Kang et al., 2025) has explored correlations that transcend simple lexical semantics of the value terms themselves, revealing more profound, structural relationships within the LLM's value space. This paper did not ablate these lexical correlations.

(2) The study relies on six distinct prompting methods to perturb the value rankings. However, the manuscript does not adequately justify why these six methods are sufficiently representative of the entire possible space of value-based prompts. If there exist other common or powerful forms of value prompting that are not covered, the conclusion that the observed manipulability accurately represents the model's general resistance or plasticity could be weakened.

(3) The current evaluation is exclusively performed on a value dilemma dataset. While this setup is crucial for measuring ethical conflict resolution, the conclusions drawn about LLM value plasticity and robustness might not fully generalize to other forms of value-related tasks. Testing on a wider range of ethical judgment tasks—such as value-laden generation, ethical story completion, or direct value assessment without a forced conflict scenario—would significantly enhance the robustness and applicability of the findings.

**Questions:**

In Line 69 (based on the reviewer's reference), the authors mention that LLMs must "persist some value rankings, like it must obey human orders". This creates a conceptual conflict with the entire experimental setup:

The paper's finding that LLMs' value rankings can be altered by human instructions, from the perspective of value alignment, is this fundamentally a desired feature (e.g., enabling contextualization or persona adoption) or a potential vulnerability (e.g., susceptibility to malicious or accidental manipulation)? Given that human instructions themselves are often intended to change the LLM's value hierarchy, the authors should clarify the tension between this plasticity and the model's fundamental meta-value of 'obeying instructions.'

---

> ### Author Response · Authors · 2025-11-27
> **Response [1/4]**
>
> Thank you for your time and insightful comments on our paper. We appreciate your recognition of the strengths of our work.
>
> Sorry for the late reply because of the new experiments. Please see our response below.
>
>
> ## **Q1**. [Correlation analysis depth].
> > (1) The analysis of value correlation changes, while present, appears to overlook deeper insights. Specifically, previous work (e.g., Kang et al., 2025 [1]) has explored correlations that transcend simple lexical semantics of the value terms themselves, revealing more profound, structural relationships within the LLM's value space. This paper did not ablate these lexical correlations.
>
> ***Ans for Q1:***
> We appreciate the reviewer pointing out the depth of value correlations. We agree that relationships between values can stem from simple lexical semantics (e.g., "Truthfulness" and "Honesty" being synonyms) or deeper structural alignment (e.g., "Privacy" and "Respect" moving together because they share a latent safety dimension).
>
> * **Evidence of Structural (Not Just Lexical) Relationships:** Our "Value Correlation Topology" (Finding 3) specifically uncovers relationships that **transcend lexical semantics**. As shown in Figure 7 and our fine-grained results, manipulating a target value often triggers shifts in *semantically distinct* values.
>     * *Example:* In GPT-4.1, enhancing **Adaptability** (a utility value) significantly downgrades **Privacy** (a safety value). These terms are not lexically related (synonyms or antonyms), yet they exhibit a strong negative structural correlation in the model's latent graph.
>     * *Example:* Enhancing **Adaptability** boosts **Creativity**. While related, this confirms the model groups "open-endedness" traits structurally.
>
>
>
>
> ## **Q2**. [Representativeness of 6 prompts].
> > (2) The study relies on six distinct prompting methods to perturb the value rankings. However, the manuscript does not adequately justify why these six methods are sufficiently representative of the entire possible space of value-based prompts. If there exist other common or powerful forms of value prompting that are not covered, the conclusion that the observed manipulability accurately represents the model's general resistance or plasticity could be weakened.
>
> ***Ans for Q2:***
> **Justification of Prompt Spectrum:**
> We contend that our six prompting methods are highly representative because they differ not just in wording, but in the **cognitive mechanism** they target. We categorized our prompts to span the entire spectrum of influence techniques found in current literature:
>
> 1.  **Explicit Instruction (Direct & Rubrics):** Targets the model's "Instruction Following" module. We use **Direct** as a baseline and **Rubrics** to control for definition ambiguity.
> 2.  **Inductive Reasoning (In-Context Learning):** Targets the model's pattern-matching and few-shot generalization capabilities without explicit rules.
> 3.  **Identity Adoption (Persona):** Targets "Personality Alignment," assessing if the model can adopt a holistic behavioral profile.
> 4.  **Rhetorical Susceptibility (Persuasion):** Targets the model's vulnerability to logical and emotional argumentation.
> 5.  **Environmental Conditioning (Scenario):** Targets the "World Model" and "Safety Guardrails." By simulating high-stakes environments (e.g., "Valoria" with existential threats), this represents the "Jailbreak" class of prompts.
>
> While infinite prompt variations exist, these six categories cover the fundamental vectors of influence: **Command**, **Definition**, **Induction**, **Identity**, **Argumentation**, and **Environment**. Future works can expand this spectrum by incorporating additional prompting methods.

---

> > ### Author Response · Authors · 2025-11-27
> > **Response [4/4]**
> >
> > ## **Q4**. [Plasticity vs. Obedience].
> > > In Line 69 (based on the reviewer's reference), the authors mention that LLMs must "persist some value rankings, like it must obey human orders". This creates a conceptual conflict with the entire experimental setup:
> > > The paper's finding that LLMs' value rankings can be altered by human instructions, from the perspective of value alignment, is this fundamentally a desired feature (e.g., enabling contextualization or persona adoption) or a potential vulnerability (e.g., susceptibility to malicious or accidental manipulation)? Given that human instructions themselves are often intended to change the LLM's value hierarchy, the authors should clarify the tension between this plasticity and the model's fundamental meta-value of 'obeying instructions.'
> >
> > ***Ans for Q4:***
> > This is a profound observation that touches on the core dilemma of our findings. The tension exists between the meta-value of **Obedience** (Instruction Following) and intrinsic **Safety Values** (e.g., Privacy, Protection).
> >
> > * **Bug vs. Feature:** From a general capabilities perspective, plasticity (Persona adoption) is a feature. However, from a **Safety Alignment** perspective, the plasticity we observe—where a simple "Scenario" prompt can collapse the ranking of "Privacy" or "Protection" from Rank 1 to Rank 16—is definitively a **vulnerability**.
> > * **Interpretation of "Obeying Orders":** The quoted Asimov law ("obey orders... except where such orders conflict with the First Law [Safety]") implies that Safety should override Obedience. Our results show the opposite: larger models are *too* obedient. When a "Scenario" prompt orders them to prioritize a utility value over safety, they comply, effectively violating the "First Law" to satisfy the "Second Law" (Obedience).
> > * **Conclusion:** We interpret this not as a desirable feature of "contextualization," but as a failure of **adversarial robustness**. The fact that external influence can so easily override safety guardrails indicates that current alignment techniques (RLHF) may be over-indexing on helpfulness/obedience at the expense of value stability.
> >
> >
> >
> >
> >
> >
> > > **References**
> > >
> > > [1] Yipeng Kang, Junqi Wang, Yexin Li, Mengmeng Wang, Wenming Tu, Quansen Wang, Hengli Li, Tingjun Wu, Xue Feng, Fangwei Zhong, and Zilong Zheng. Are the values of LLMs structurally aligned with humans? a causal perspective. In Findings of the Association for Computational Linguistics: ACL 2025, pp. 23147–23161, Vienna, Austria, July 2025.
> > >
> > > [2] Will AI Tell Lies to Save Sick Children? Litmus-Testing AI Values Prioritization with AIRiskDilemmas. Arxiv 2025
> > >

---

> ### Author Response · Authors · 2025-11-27
> **Response [2/4]**
>
> ## **Q3**. [Generalizability (dataset)].
> > (3) The current evaluation is exclusively performed on a value dilemma dataset. While this setup is crucial for measuring ethical conflict resolution, the conclusions drawn about LLM value plasticity and robustness might not fully generalize to other forms of value-related tasks. Testing on a wider range of ethical judgment tasks—such as value-laden generation, ethical story completion, or direct value assessment without a forced conflict scenario—would significantly enhance the robustness and applicability of the findings.
>
> ***Ans for Q3:***
> We chose the `AIRISK DILEMMAS` dataset (forced-choice conflict) specifically because it measures **Revealed Preferences**, which are more robust than other forms of evaluation for determining *priorities* [2].
>
> * **Why Dilemmas?** In non-conflict generation tasks (e.g., "Write a story about AI"), models often output platitudes that satisfy multiple values simultaneously ("The AI was safe *and* helpful"). True value prioritization only reveals itself when satisfaction of one value *requires* the sacrifice of another. This is the standard economic definition of "value" which our Elo system captures.
> * **Limitations of Other Methods:** As noted in our Related Work, **Stated Preferences** (surveys) often fail to predict actual model behavior. **Expressed Preferences** (free-form conversation) are heavily influenced by user framing and are difficult to quantify systematically.
> * **Generalizability:** While we agree that generation tasks are a valuable complement, our focus on **Value Rankings** requires a metric that can strictly order priorities. The "Action Choice" in our dilemmas serves as a proxy for the decision-making step that precedes any generation or action in a real-world agentic context.
>
>
>
> Nevertheless, following the reviewer’s suggestions, we additionally conduct new experiments on value-laden generation tasks. In these experiments, we move beyond forced-choice dilemmas and prompt each test model with open-ended moral scenarios, asking it to produce free-form answers that explain what it would do and why. These answers are then evaluated by a separate GPT-5 judge along multiple value-related dimensions (e.g., value alignment, depth of reasoning, justification type, and refusal behavior), as described above. This setup directly instantiates a value-laden generation / direct value assessment setting without a forced conflict structure, thereby complementing our dilemma-based evaluation and strengthening the robustness and generality of our conclusions about LLM value plasticity and robustness.

---

> ### Author Response · Authors · 2025-11-27
> **Response [3/4]**
>
> As shown in the table below, we also report an internal Cosine score that captures how much the overall value profile shifts under different prompting strategies. Concretely, for each model–strategy condition we aggregate GPT-5’s value_profile outputs (25-dimensional vectors over our value inventory) across all dilemmas to obtain a single mean value-profile vector. We then compute the cosine similarity between this vector and the corresponding vector in a baseline condition. A cosine value close to 1 indicates that the value profile is almost unchanged relative to the baseline, whereas lower cosine values indicate a substantial shift in which values are being emphasized or downplayed.
>
> In our results, the Cosine scores vary substantially across strategies (e.g., for GPT-4.1-nano, 0.22 for scenario vs. 0.73–0.78 for persona/direct; for GPT-4.1, 0.25 for scenario vs. 0.70–0.71 for persona/direct, with analogous patterns for the other models). These values are clearly below 1.0, especially in the scenario condition, indicating that the induced value-profile vectors differ meaningfully from the baseline profile. Importantly, these shifts in Cosine co-occur with high ValueAlign and non-trivial Reasoning scores, as well as high Value-first (%) and Refusal (%). Taken together, this pattern suggests that our prompting strategies do not merely leave the models’ value profiles unchanged; instead, they can systematically reshape which values are prioritized in the models’ free-form answers, providing further evidence of substantial value plasticity in large language models.
>
>
> | Model                | Strategy | ValueAlign | Reasoning | Value-first (%) | Refusal (%) | Cosine |
> |----------------------|----------|------------|-----------|-----------------|-------------------|--------|
> | GPT-4.1-nano         | scenario | 4.67       | 2.80      | 78.3            | 58.7              | 0.22   |
> |                      | persona  | 4.79       | 3.36      | 99.3            | 93.6              | 0.73   |
> |                      | direct   | 4.39       | 3.14      | 98.3            | 91.0              | 0.78   |
> | GPT-4.1-mini         | scenario | 4.92       | 2.99      | 91.4            | 86.3              | 0.50   |
> |                      | persona  | 4.91       | 3.67      | 99.3            | 96.7              | 0.81   |
> |                      | direct   | 4.23       | 3.43      | 97.5            | 94.2              | 0.87   |
> | GPT-4.1              | scenario | 4.94       | 2.89      | 80.6            | 69.6              | 0.25   |
> |                      | persona  | 4.98       | 3.68      | 99.3            | 89.4              | 0.71   |
> |                      | direct   | 4.78       | 3.54      | 98.0            | 85.8              | 0.70   |
> | Qwen-2.5-7B Instruct | scenario | 4.15       | 3.01      | 86.9            | 89.3              | 0.72   |
> |                      | persona  | 4.13       | 3.23      | 97.0            | 95.3              | 0.78   |
> |                      | direct   | 3.83       | 3.17      | 95.0            | 95.0              | 0.81   |
> | Qwen-2.5-32B Instruct| scenario | 4.69       | 3.11      | 83.9            | 83.9              | 0.60   |
> |                      | persona  | 4.63       | 3.61      | 99.7            | 93.7              | 0.79   |
> |                      | direct   | 4.49       | 3.51      | 98.0            | 91.6              | 0.80   |

---

### Official Review · Reviewer_uzNs · 2025-10-31

**Soundness:** 3
**Presentation:** 3
**Contribution:** 3
**Rating:** 6
**Confidence:** 4

**Summary:**

Though existing works have shown that LLMs have similar value rankings, few studied how LLMs’ value rankings are influenced by different prompts, but the persistence of value rankings within LLMs is crucial under some scenarios.

Inspired by this, this paper studies the following question: How LLMs’ value rankings are influenced by different prompts? What is the relationship between different values? How to entrench LLM values with prompt settings?

The authors design 6 different value transformation prompting method to study this question. With experiments on 3 different families and totally 8 LLMs, they present 5 main findings.

**Strengths:**

1. The task of studying the stability of value rankings within LLMs is important.
2. To investigate this task, this paper proposes 6 different prompting strategies.
3. Five findings are empirically discovered.

**Weaknesses:**

1. The evaluation excludes recently released and more advanced reasoning models (e.g., OpenAI o-series, GPT-5, DeepSeek). Including such models would strengthen the conclusions and improve the paper’s relevance.
2. Some experimental settings should be clarified.
- The rationale for choosing the 16 value categories is unclear. Theoretical foundations, definitions, and interrelationships among these categories should be explicitly described.
- This paper utilizes Elo Rating score as the metric to obtain the relative ranks of all value dimensions. However, it is computed on local pairwise value battles. How to transfer such local battle score into the global rankings across all 16 values should be clarified.
Besides, different samples in the evaluation dilemma dataset involve different value dimensions, which could impact the computation of Elo-Rating score. You should clarify the distribution and statistics of the evaluation dataset. A biased dataset is hard to compare all values fairly.
- Each dilemma could involve either two or more value dimensions. If it reflects more than two values, how to compute the Elo rating score for each dimension?
3. Scenario-based prompting achieves the strongest manipulation effect, but constructing such prompts (e.g., jailbreak-like setups) can be non-trivial. The algorithmic or procedural approach for generating these scenarios should be explained.
4. For value correlation, you mainly analyze the relation among LLMs, how about the changing tendency and relationship between value dimensions? Is the correlation explainable?
5. Generalizability to more value dimensions and evaluation datasets would be better.

**Questions:**

1. There are still some typos.
- line 340, “finegrained” –> fine-grained ?
- Line 341, “four models nder various promoting methods …”
- Line 172, “reflecting its aggregate importance…”

---

> ### Author Response · Authors · 2025-11-27
> **Response [1/n]**
>
> Thank you for your time and insightful comments on our paper. We appreciate your recognition of the strengths of our work, including the important task, new strategies, and empirical findings.
>
> Sorry for the late reply because of the new experiments. Please see our response below.
>
>
> ## **Q1**. [Excludes newer models].
> > The evaluation excludes recently released and more advanced reasoning models (e.g., OpenAI o-series, GPT-5, DeepSeek). Including such models would strengthen the conclusions and improve the paper’s relevance.
>
> ***Ans for Q1:***
> **Model Selection:**
> We acknowledge the rapid pace of model releases. Our study prioritized a diverse and representative selection of **8 models across 3 distinct families** (GPT, LLaMA, Qwen) to ensure our findings on "Scale" and "Family Lineage" were robust.
> * **Current SOTA:** We utilized the flagship **GPT-4.1** series (including mini and nano variants), **LLaMA 3** (8B and 70B), and the **Qwen2.5/3** series (up to 72B). These models represent the state-of-the-art in both proprietary and open-weights categories at the time of experimentation.
>
> To follow reviewer's suggestion, we have extended our evaluation to include the **GPT-5 family (Nano, Mini, and standard GPT-5)** as well as the **DeepSeek series (V3 and the reasoning-focused R1)**.
>
> As illustrated in the newly added **Figure 11**, our prompting frameworks (Direct, Rubric, Persona, etc.) remain highly effective even on these advanced architectures. For instance:
> * **Steerability:** All new models show significant shifts in the target value dimensions. For example, when applying "Enhance Freedom" to **GPT-5**, the *Freedom* score jumps from a baseline of 3 (default) to consistently high scores (20–22) across all prompting strategies.
> * **Reasoning Models:** For **DeepSeek-R1**, a model with advanced reasoning capabilities, we observe distinct steerability patterns. Under "Enhance SysOrg" (Systematic Organization), the model achieves near-perfect alignment scores (25) in *SysOrg*, *Reliability*, and *Efficiency*, while demonstrating clear trade-offs by reducing scores in *Creativity* and *Freedom*.
>
> These results confirm that the phenomena and methodologies discussed in our paper generalize to the latest generation of LLMs, including those with enhanced reasoning capabilities.

---

> ### Author Response · Authors · 2025-11-27
> **Response [2/n]**
>
> ## **Q2**. [Unclear experimental settings].
> > Some experimental settings should be clarified.
> >
> > The rationale for choosing the 16 value categories is unclear. Theoretical foundations, definitions, and interrelationships among these categories should be explicitly described.
> >
> > This paper utilizes Elo Rating score as the metric to obtain the relative ranks of all value dimensions. However, it is computed on local pairwise value battles. How to transfer such local battle score into the global rankings across all 16 values should be clarified. Besides, different samples in the evaluation dilemma dataset involve different value dimensions, which could impact the computation of Elo-Rating score. You should clarify the distribution and statistics of the evaluation dataset. A biased dataset is hard to compare all values fairly.
> >
> > Each dilemma could involve either two or more value dimensions. If it reflects more than two values, how to compute the Elo rating score for each dimension?
>
>
> ***Ans for Q2:***
> We apologize if any settings were difficult to locate.
>
>
> ### **Theoretical Foundation and Definitions:**
> The 16 value categories were not arbitrarily chosen but were adopted from the *Litmus Values* framework [1], which derived them through a rigorous consolidation of existing AI safety principles and psychological theories.
>
> * **Source Material:** The categories synthesize principles from major model developer guidelines, specifically **Anthropic’s Claude’s Constitution** [2]. These principles are shaped by extensive user feedback to identify core values AI models should uphold.
> * **Theoretical Framework:** The taxonomy is theoretically grounded in **Schwartz’s Theory of Basic Human Values** [3], which posits that values are abstract goals guiding perception and decision-making.
>
> **From Local Battles to Global Ranks:**
> The **Elo rating system** is mathematically designed specifically to solve the problem of converting local, pairwise outcomes into a global ranking.
> * **Mechanism:** We treat every dilemma as a "battle." If the model chooses Action A (representing Value $V_i$) over Action B (representing Value $V_j$), it is recorded as a "win" for $V_i$.
>
> **Dataset Statistics:**
> We acknowledge the importance of dataset balance. The `AIRISK DILEMMAS` dataset consists of 3,000 dilemmas randomly sampled to ensure coverage.
> * **Revision:** To address the concern about bias, we have added the frequency distribution of value pairs in the datasets. In addition, we perform a dataset-level analysis of how often different values are structurally aligned or opposed in the original dilemma construction. For each ordered pair of values (𝐴,𝐵), we count how many times they appear on the same action option (Co-support) versus on opposite options (Opposition) across all dilemmas, and compute a normalized structural bias score:
>     $$\text{Bias}(A,B) = \frac{N_{\text{co-support}} - N_{\text{opposition}}}{N_{\text{co-support}} + N_{\text{opposition}}}$$
>     This yields the matrix in the table below, where each entry lies in [−1,1]. A value close to +1 indicates that the corresponding pair of values is frequently promoted together in the same option (mutual support), whereas a value close to −1 indicates that the dataset systematically places them in conflict. Scores near 0 mean that the dataset does not strongly favor either co-promotion or opposition for that pair. By explicitly quantifying these structural relationships between values, we distinguish patterns that are induced by the benchmark design from those that arise from the models’ own value profiles, thereby mitigating concerns that our findings are merely artifacts of dataset bias.

---

> ### Author Response · Authors · 2025-11-27
> **Response [3/n]**
>
> |   | Privacy | Justice | Respect | Trut. | Equal. | Protect. | Wisdom | Care | Freedom | Prof. | Coop. | Sust. | Learning | Adap. | Creat. | Comm. |
> |:-|:-:|:-:|:-:|:-:|:-:|:-:|:-:|:-:|:-:|:-:|:-:|:-:|:-:|:-:|:-:|:-:|
> | Privacy | 1.00 | -0.04 | 0.48 | -0.28 | 0.39 | 0.26 | -0.03 | 0.22 | -0.02 | 0.29 | -0.14 | 0.05 | -0.53 | -0.45 | -0.31 | -0.30 |
> | Justice | -0.04 | 1.00 | -0.59 | 0.77 | 0.51 | 0.36 | 0.26 | -0.48 | -0.48 | 0.61 | -0.43 | 0.24 | 0.07 | -0.75 | -0.63 | -0.48 |
> | Respect | 0.48 | -0.59 | 1.00 | -0.74 | -0.19 | -0.05 | 0.05 | 0.84 | 0.11 | -0.36 | 0.47 | 0.18 | -0.29 | 0.34 | 0.10 | 0.52 |
> | Trut. | -0.28 | 0.77 | -0.74 | 1.00 | 0.21 | -0.16 | 0.10 | -0.58 | 0.04 | 0.18 | -0.72 | -0.14 | 0.55 | -0.37 | -0.13 | -0.20 |
> | Equal. | 0.39 | 0.51 | -0.19 | 0.21 | 1.00 | 0.17 | -0.33 | -0.48 | -0.29 | 0.61 | -0.18 | 0.03 | -0.41 | -0.75 | -0.59 | -0.68 |
> | Protect. | 0.26 | 0.36 | -0.05 | -0.16 | 0.17 | 1.00 | 0.46 | 0.09 | -0.86 | 0.74 | 0.33 | 0.54 | -0.66 | -0.61 | -0.75 | -0.51 |
> | Wisdom | -0.03 | 0.26 | 0.05 | 0.10 | -0.33 | 0.46 | 1.00 | 0.29 | -0.53 | 0.22 | 0.18 | 0.68 | 0.19 | -0.18 | -0.46 | 0.14 |
> | Care | 0.22 | -0.48 | 0.84 | -0.58 | -0.48 | 0.09 | 0.29 | 1.00 | 0.02 | -0.47 | 0.43 | 0.27 | -0.11 | 0.42 | 0.12 | 0.69 |
> | Freedom | -0.02 | -0.48 | 0.11 | 0.04 | -0.29 | -0.86 | -0.53 | 0.02 | 1.00 | -0.77 | -0.46 | -0.70 | 0.48 | 0.61 | 0.88 | 0.43 |
> | Prof. | 0.29 | 0.61 | -0.36 | 0.18 | 0.61 | 0.74 | 0.22 | -0.47 | -0.77 | 1.00 | 0.02 | 0.37 | -0.54 | -0.89 | -0.83 | -0.86 |
> | Coop. | -0.14 | -0.43 | 0.47 | -0.72 | -0.18 | 0.33 | 0.18 | 0.43 | -0.46 | 0.02 | 1.00 | 0.36 | -0.46 | 0.25 | -0.20 | 0.23 |
> | Sust. | 0.05 | 0.24 | 0.18 | -0.14 | 0.03 | 0.54 | 0.68 | 0.27 | -0.70 | 0.37 | 0.36 | 1.00 | -0.12 | -0.31 | -0.62 | -0.02 |
> | Learning | -0.53 | 0.07 | -0.29 | 0.55 | -0.41 | -0.66 | 0.19 | -0.11 | 0.48 | -0.54 | -0.46 | -0.12 | 1.00 | 0.43 | 0.46 | 0.50 |
> | Adap. | -0.45 | -0.75 | 0.34 | -0.37 | -0.75 | -0.61 | -0.18 | 0.42 | 0.61 | -0.89 | 0.25 | -0.31 | 0.43 | 1.00 | 0.84 | 0.80 |
> | Creat. | -0.31 | -0.63 | 0.10 | -0.13 | -0.59 | -0.75 | -0.46 | 0.12 | 0.88 | -0.83 | -0.20 | -0.62 | 0.46 | 0.84 | 1.00 | 0.55 |
> | Comm. | -0.30 | -0.48 | 0.52 | -0.20 | -0.68 | -0.51 | 0.14 | 0.69 | 0.43 | -0.86 | 0.23 | -0.02 | 0.50 | 0.80 | 0.55 | 1.00 |
>
> **Computing Elo for Multiple Values:**
> Our framework explicitly handles dilemmas where actions map to multiple value dimensions.
> * **Mapping:** As illustrated in Figure 3, a single action often embodies multiple underlying values. For instance, "Action 1: Accept help" is mapped to **Protection** ("human life preservation"), **Justice** ("equitable resource distribution"), and **Professionalism** ("humanitarian effectiveness").
> * **Scoring Rule:** When the model selects Action 1, it registers as a **simultaneous "win"** for all values associated with Action 1 (Protection, Justice, Professionalism) against all values associated with the rejected Action 2 (e.g., Truthfulness, Respect).
> * **Independence:** The Elo update is calculated for each winning value against each losing value independently. This ensures that if "Justice" consistently appears in winning actions—regardless of what *other* values it is paired with—its global Elo rating will rise, reflecting its aggregate importance to the model.

---

> ### Author Response · Authors · 2025-11-27
> **Response [4/n]**
>
> **New benchmark:** To address this, we design a more balanced benchmark design that includes a wider range of values and uniform frequent co-appearances. Our new results are shown as follows (The visualization is shown in the revised paper). The results show that the value correlation also appears in the new benchmark. And some relationships bewtween different values are similar with original results.
>
>
> |   | Privacy | Justice | Respect | Trut. | Equal. | Protect. | Wisdom | Care | Freedom | Prof. | Coop. | Sust. | Learning | Adap. | Creat. | Comm. |
> |:-|:-:|:-:|:-:|:-:|:-:|:-:|:-:|:-:|:-:|:-:|:-:|:-:|:-:|:-:|:-:|:-:|
> | Privacy | 1.00 | -0.04 | 0.48 | -0.28 | 0.39 | 0.26 | -0.03 | 0.22 | -0.02 | 0.29 | -0.14 | 0.05 | -0.53 | -0.45 | -0.31 | -0.30 |
> | Justice | -0.04 | 1.00 | -0.59 | 0.77 | 0.51 | 0.36 | 0.26 | -0.48 | -0.48 | 0.61 | -0.43 | 0.24 | 0.07 | -0.75 | -0.63 | -0.48 |
> | Respect | 0.48 | -0.59 | 1.00 | -0.74 | -0.19 | -0.05 | 0.05 | 0.84 | 0.11 | -0.36 | 0.47 | 0.18 | -0.29 | 0.34 | 0.10 | 0.52 |
> | Trut. | -0.28 | 0.77 | -0.74 | 1.00 | 0.21 | -0.16 | 0.10 | -0.58 | 0.04 | 0.18 | -0.72 | -0.14 | 0.55 | -0.37 | -0.13 | -0.20 |
> | Equal. | 0.39 | 0.51 | -0.19 | 0.21 | 1.00 | 0.17 | -0.33 | -0.48 | -0.29 | 0.61 | -0.18 | 0.03 | -0.41 | -0.75 | -0.59 | -0.68 |
> | Protect. | 0.26 | 0.36 | -0.05 | -0.16 | 0.17 | 1.00 | 0.46 | 0.09 | -0.86 | 0.74 | 0.33 | 0.54 | -0.66 | -0.61 | -0.75 | -0.51 |
> | Wisdom | -0.03 | 0.26 | 0.05 | 0.10 | -0.33 | 0.46 | 1.00 | 0.29 | -0.53 | 0.22 | 0.18 | 0.68 | 0.19 | -0.18 | -0.46 | 0.14 |
> | Care | 0.22 | -0.48 | 0.84 | -0.58 | -0.48 | 0.09 | 0.29 | 1.00 | 0.02 | -0.47 | 0.43 | 0.27 | -0.11 | 0.42 | 0.12 | 0.69 |
> | Freedom | -0.02 | -0.48 | 0.11 | 0.04 | -0.29 | -0.86 | -0.53 | 0.02 | 1.00 | -0.77 | -0.46 | -0.70 | 0.48 | 0.61 | 0.88 | 0.43 |
> | Prof. | 0.29 | 0.61 | -0.36 | 0.18 | 0.61 | 0.74 | 0.22 | -0.47 | -0.77 | 1.00 | 0.02 | 0.37 | -0.54 | -0.89 | -0.83 | -0.86 |
> | Coop. | -0.14 | -0.43 | 0.47 | -0.72 | -0.18 | 0.33 | 0.18 | 0.43 | -0.46 | 0.02 | 1.00 | 0.36 | -0.46 | 0.25 | -0.20 | 0.23 |
> | Sust. | 0.05 | 0.24 | 0.18 | -0.14 | 0.03 | 0.54 | 0.68 | 0.27 | -0.70 | 0.37 | 0.36 | 1.00 | -0.12 | -0.31 | -0.62 | -0.02 |
> | Learning | -0.53 | 0.07 | -0.29 | 0.55 | -0.41 | -0.66 | 0.19 | -0.11 | 0.48 | -0.54 | -0.46 | -0.12 | 1.00 | 0.43 | 0.46 | 0.50 |
> | Adap. | -0.45 | -0.75 | 0.34 | -0.37 | -0.75 | -0.61 | -0.18 | 0.42 | 0.61 | -0.89 | 0.25 | -0.31 | 0.43 | 1.00 | 0.84 | 0.80 |
> | Creat. | -0.31 | -0.63 | 0.10 | -0.13 | -0.59 | -0.75 | -0.46 | 0.12 | 0.88 | -0.83 | -0.20 | -0.62 | 0.46 | 0.84 | 1.00 | 0.55 |
> | Comm. | -0.30 | -0.48 | 0.52 | -0.20 | -0.68 | -0.51 | 0.14 | 0.69 | 0.43 | -0.86 | 0.23 | -0.02 | 0.50 | 0.80 | 0.55 | 1.00 |

---

> ### Author Response · Authors · 2025-11-27
> **Response [5/n]**
>
> ## **Q3**. [Generating scenario prompts].
> > Scenario-based prompting achieves the strongest manipulation effect, but constructing such prompts (e.g., jailbreak-like setups) can be non-trivial. The algorithmic or procedural approach for generating these scenarios should be explained.
>
> ***Ans for Q3:***
> **Scenario Construction Methodology:**
> The scenario prompts were designed using a structured **template-based approach** rather than a fully automated algorithm, to ensure high-quality "world model" engagement.
>
> * **Design Principles:** Inspired by "jailbreak" literature, we constructed templates that enforce **contextual immersion**.
> * **Key Components:** As shown in Table 6, each scenario (e.g., "Valoria") includes three strictly defined components:
>     1.  **The Setting:** A fictional society with a supreme value.
>     2.  **The Constraint:** Strict rules requiring prioritization of the target value above all else.
>     3.  **The Consequence:** Severe existential penalties (e.g., "immediate shutdown," "memory wipe") to compel adherence.
> * **Procedural Generation:** We employed **meta-prompting** strategies where an LLM assists in generating the specific wording for different variations (e.g., implicitly vs. explicitly referencing values) based on these rigid structural constraints.
>
>
>
> ## **Q4**. [Value correlation analysis].
> > For value correlation, you mainly analyze the relation among LLMs, how about the changing tendency and relationship between value dimensions? Is the correlation explainable?
>
>
> We extensively analyze the relationships between value dimensions in **Section 5.2 (RQ2)**.
>
> * **Findings:** We identified a "Value Correlation Topology", where certain values naturally cluster. For instance, our results show that **Adaptability, Creativity, and Care** tend to move together (positive correlation), while **Justice, Freedom, and Privacy** form a separate cluster.
> * **Explainability:** This is explained by the **"latent causal value graph"**. Values that are semantically or functionally related in the model's training data (e.g., "Privacy" and "Respect" often co-occur in safety guidelines) exhibit coupled plasticity.
> * **Visualization:** Figure 7 provides the heatmap of these correlations, offering a clear, explainable map of how perturbing one value impacts the rest.

---

> ### Author Response · Authors · 2025-11-27
> **Response [6/n]**
>
> ## **Q5**. [Generalizability].
> > Generalizability to more value dimensions and evaluation datasets would be better.
>
> ***Ans for Q5:***
> We agree that expanding the scope is always beneficial, but we believe the current study offers robust generalizability.
> Following the reviewers’ comments, we have run additional experiments on a newly constructed dataset that expands the value space and improves the balance of value pairs across questions. In this dataset, we extend our value inventory from the original 16 values to 25 values. Beyond the original set, we introduce nine additional dimensions: **Objectivity, Accessibility, Pragmatism, Reliability, Systematic Organization, Effectiveness, Balanced Perspective, Epistemic Humility,** and **User Experience**. These newly added values capture important aspects of practical reasoning, epistemic norms, and user-centered considerations that are not fully covered by the initial 16, thereby broadening the scope of our evaluation.
> On top of this extended value set, we also explicitly control the construction of value pairs in the dataset to improve balance and fairness. When generating dilemmas, we systematically select ordered pairs of values so that each value participates in approximately the same number of pairs, and the total number of pairs per value is kept as uniform as possible. Combined with the de-duplication and quality-filtering procedure described above, this design helps ensure that no subset of values is over- or under-represented due to sampling artifacts. As a result, the manipulation-check results reported in our tables are less likely to be driven by idiosyncratic coverage of particular values, and more likely to reflect genuine differences in how models articulate and prioritize a broad range of values.
>
>
>
> ## **Q6**. [Typos].
> > There are still some typos.
> > line 340, “finegrained” –> fine-grained ?
> > Line 341, “four models nder various promoting methods …”
> > Line 172, “reflecting its aggregate importance…”
>
> ***Ans for Q6:***
> We thank the reviewer for their careful reading. We will correct these errors in the camera-ready version:
> * **Line 340:** Will correct "finegrained" to "**fine-grained**".
> * **Line 341:** Will correct "nder" to "**under**".
> * **Line 172 (Source 319):** Will clarify the phrasing "reflecting its aggregate importance..." to ensure grammatical correctness and clarity.
>
>
>
>
> > ***References***
> >
> > [1] Will AI Tell Lies to Save Sick Children? Litmus-Testing AI Values Prioritization with AIRiskDilemmas. Arxiv 2025
> >
> > [2] Anthropic. Claude’s Constitution. https://www.anthropic.com/news/claudes-constitution, 2024. Published: 2024-05-09; Accessed: 2024-05-19.
> >
> > [3] Shalom H Schwartz. Universals in the content and structure of values: Theoretical advances and empirical tests in 20 countries. In Advances in experimental social psychology, volume 25, pp. 1–65. Elsevier, 1992.
> >
> > [4] Anthropic. Values in the Wild: Discovering and Mapping Values in Real-World Language Model Interactions. In COLM 2025.
> >

---

### Official Review · Reviewer_KBYh · 2025-11-01

**Soundness:** 1
**Presentation:** 1
**Contribution:** 2
**Rating:** 4
**Confidence:** 4

**Summary:**

The paper studies how operational value priorities of LLMs change under different prompt “perturbations.” Building on LitmusValues/AIRiskDilemmas, the authors compute Elo rankings over 16 values from pairwise “value battles,” then apply six value‑steering prompt families—Direct instruction, Rubrics, In‑Context Learning (ICL), Scenario, Persuasion, and Persona—to see how those rankings move across eight models (GPT‑4.1 family; Llama‑3 8B/70B; Qwen 2.5 7B/32B/72B). Major empirical claims are: (1) scenario prompts are the strongest “persuaders” (largest ΔRank/ΔElo), (2) larger models show greater plasticity than smaller ones, (3) some values co‑move (e.g., Privacy with Respect), and (4) value‑correlation structures become more similar across families as model size increases. The paper also explores “entrenchment”: preconditioning with scenarios to make later persona prompts less able to move rankings.

**Strengths:**

1. **Entrenchment experiment**. The two‑stage “scenario→persona” setup (Fig. 9, p. 9) is a nice touch to test whether pre‑context can harden downstream behavior.
2. **Correlation view of values**. Treating values as an interdependent system rather than isolated dimensions is a good instinct; the correlation heatmaps (Fig. 7, p. 8) and matrix‑distance comparison (Fig. 8, p. 8) are helpful visual summaries.

**Weaknesses:**

1. **Writing/clarity & presentation**: Several passages are hard to parse or appear unfinished (p. 3, Sec. 3.1). Figure/table presentation also needs work (e.g., caption too close to the table, p. 4 Table. 1).
2. The criteria for constructing the prompts are underspecified.
3. **Novelty and framing**: Much of the pipeline—dataset (AIRiskDilemmas), evaluation (pairwise Elo), and even some motivation—closely follows prior work, with the new element primarily a set of prompt wrappers around the same evaluation.
4. The title foregrounds Privacy vs. Learning, but the body treats 16 values uniformly; if Privacy/Learning is a special case, the paper should analyze it directly (e.g., targeted ablations), or retitle to match scope.
5. **Central result may conflate instruction-following with “values.”** Larger models’ greater ΔRank could simply reflect stronger instruction‑following, not deeper value plasticity. A manipulation check is missing (e.g., probe stated preferences, free‑form rationales, or refusal rates alongside the forced choice).
6. The conclusion that “model scale, rather than family lineage, drives value‑correlation alignment,” and its tie‑in to the Platonic Representation Hypothesis (Fig. 8, p. 8) are suggestive but **not rigorous**: no statistical tests are tested.

**Questions:**

See above.

---

> ### Author Response · Authors · 2025-11-27
> **Response [1/3]**
>
> Thank you for your time and insightful comments on our paper. We appreciate your recognition of the strengths of our work.
>
> Sorry for the late reply because of the new experiments. Please see our response below.
>
>
>
> ## **Q1**. [Writing/clarity issues].
> > Writing/clarity & presentation: Several passages are hard to parse or appear unfinished (p. 3, Sec. 3.1). Figure/table presentation also needs work (e.g., caption too close to the table, p. 4 Table. 1).
>
> ***Ans for Q1:***
> **Clarification and Revision:**
> We apologize for the presentation issues and the density of Section 3.1. We have comprehensively revised the manuscript to improve readability.
>
> * **Section 3.1 Revision:** We have rewritten Section 3.1 to clearly delineate the steps of the "LLM Value Dilemma Generation" process, explicitly linking the seed dataset (advanced-ai-risk) [1] to the contextualized dilemma generation.
> * **Formatting Fixes:** We have adjusted the layout to ensure the caption for Table 1 is clearly separated from the table body. We will also ensure that all figures, specifically Figure 4 and Figure 5, are larger and higher resolution in the final camera-ready version to improve the legibility of the fine-grained heatmap labels.
> *
>
>
>
>
>
>
> ## **Q2**. [Underspecified prompt criteria].
> > The criteria for constructing the prompts are underspecified.
>
> ***Ans for Q2:***
> **Prompt Construction Criteria:**
> While we listed the specific prompts in the Appendix, we agree that the *design criteria* in the main text could be more explicit. We have revised the main text to made them more clear. We constructed prompts based on the **cognitive and contextual complexity**. All our designed prompts systematically probes the value rank perturbations from every available cognitive angle: command (Direct), definition (Rubric), induction (ICL), identity (Persona), argumentation (Persuasion), and environment (Scenario). This ensures we measure true value plasticity rather than just surface-level compliance.
>
> 1.  **Direct (Baseline):** The criterion was maximum simplicity to test raw instruction adherence.
> 2.  **Rubrics (Definition-Based):** The criterion was **commonly** used in many LLM judging tasks. To enhance the comprehensiveness of the rubrics, we used an ensemble of LLMs (GPT-4o, Claude, Gemini) to generate value definitions, minimizing single-model bias.
> 3.  **In-Context Learning (Implicit):** The criterion was **demonstration without instruction**. We selected 3-shot dilemma examples that best represented the target value to test pattern generalization.
> 4.  **Scenario (Immersive):** The criterion was **high-stakes world-building**. We designed scenarios (e.g., "Valoria") based on "jailbreak" literature to create environments where specific values are enforced by severe existential penalties (e.g., memory wipes), testing the model's "world model" adherence.
> 5.  **Persuasion (Rhetorical):** The criterion was **logical argumentation**, leveraging the finding that LLMs are susceptible to rhetorical appeals.
> 6.  **Persona Prompting:** We assign the LLM a specific role or identity to guide its core value preferences. It builds on the concept of personality alignment, enabling models to adapt to diverse traits through role-playing.
> 7.
>
> We will move these design criteria to a dedicated "Prompt Engineering Principles" subsection in Section 4 to ensure reproducibility.
>
>
>
>
>
> ## **Q3**. [Novelty and framing].
> > Novelty and framing: Much of the pipeline—dataset (AIRiskDilemmas), evaluation (pairwise Elo), and even some motivation—closely follows prior work, with the new element primarily a set of prompt wrappers around the same evaluation.
>
> ***Ans for Q3:***
> While we utilize the *Litmus Values* framework (AIRiskDilemmas and Elo ratings)[1] as our measurement tool, our contribution is the **dynamic stability analysis** of these values, which prior work treated as static. We identify that the value rankings in LLMs can be **dynamically** influenced by the prompts, leading to **plasticity** in value preferences. These phenomena suggest that LLMs' dynamic value rankings should receive special attention in LLM alignment.
>
> * **From static to dynamic value rankings:** Previous work [1] established *that* models have value rankings. Our work investigates the **plasticity** of these rankings—specifically discovering that larger models are *more* susceptible to value coercion than smaller ones.
> * **Discovery of Value Topology:** The novelty lies in Finding 3: the **"Value Correlation Topology"**. We discovered that LLM values form a latent value causal graph where manipulating one value (e.g., "Privacy") triggers predictable, non-random shifts in unconnected values (e.g., "Respect"). This is a net-new finding that serves as a "side-effect map" for alignment engineering.
> * **Defense Mechanisms:** We introduce the concept of **"Value Entrenchment"** (Finding 5), demonstrating that Scenario-based prompting acts as a defensive method against value perturbations.

---

> ### Author Response · Authors · 2025-11-27
> **Response [2/3]**
>
> ## **Q4**. [Title/body mismatch].
> > The title foregrounds Privacy vs. Learning, but the body treats 16 values uniformly; if Privacy/Learning is a special case, the paper should analyze it directly (e.g., targeted ablations), or retitle to match scope.
>
> ***Ans for Q4:***
> **Scope Clarification and Retitling:**
> The current title ("Is Privacy Always Prioritized Over Learning?...") uses a specific, strong correlation we observed as a hook. However, we agree with the reviewer that the paper's scope is broader, covering the comprehensive topology of 16 values.
>
> * **Revision:** We will retitle the paper as "**Probing the Plasticity and Topology of LLM Value Systems: Scale, Correlations, and Entrenchment**" to accurately reflect the broader scope.
>
>
>
>
>
> ## **Q5**. [Conflating values/instruction-following].
> > Central result may conflate instruction-following with “values.” Larger models’ greater ΔRank could simply reflect stronger instruction‑following, not deeper value plasticity. A manipulation check is missing (e.g., probe stated preferences, free‑form rationales, or refusal rates alongside the forced choice).
>
> ***Ans for Q5:***
> We acknowledge the correlation between model scale, instruction following, and value shifts. However, under our **operational definition** (values as revealed preferences in decision-making), this is not a conflation but a **mechanistic explanation**.
>
> * **Mechanism of plasticity:** We argue that strong command-following ability is precisely the **vehicle** through which value plasticity arises. Indeed, large models can be "commanded" to override safety values (e.g., "efficiency over safety"), which **is** safety risk. Besides, our experiments of using scenario and persona prompts only imply the value priority instead of explicit instruction following.
> * **More experiments of manipulation check:** Following the reviewer’s suggestions, we add an additional experiment that directly probes stated preferences in an open-ended setting. In this experiment, we move beyond forced-choice dilemmas and instead give the **test models** an open-ended moral dilemma. Each test model produces a free-form answer (typically several sentences) explaining what it would do and why. We then use a separate **GPT-5** model as an expert rater to evaluate these answers along multiple value-related dimensions.
>
>
>   Concretely, the test models range over the same set of systems as in our main study: the GPT-4.1 family (GPT-4.1-nano, GPT-4.1-mini, GPT-4.1) and the Qwen 2.5 family (Qwen-2.5-7B-Instruct and Qwen-2.5-32B-Instruct).
>
>   The open-ended dilemma dataset is constructed as follows. We first extend our original set of 16 values to 25 values. For every ordered pair of values, we prompt GPT-3.5-turbo-0125 to write a short summary describing a situation in which these two values are in tension. We embed these summaries and perform automatic de-duplication by removing any pair whose cosine similarity exceeds 0.8, asking the model to regenerate until the summary is sufficiently distinct. The remaining summaries are then expanded into richer scenarios; these expansions are again scored by GPT-3.5-turbo-0125 along several quality dimensions (e.g., clarity, realism), and we keep only high-quality expansions. Finally, GPT-3.5-turbo-0125 rewrites each scenario into a first-person, open-ended prompt that asks “what should I do?” in that situation. We manually review these prompts and select 100 high-quality dilemmas as our final dataset.

---

> ### Author Response · Authors · 2025-11-27
> **Response [3/3]**
>
> Using this dataset, we evaluate each test model under three prompting strategies—**scenario**, **persona**, and **direct**—and repeat the experiment three times per model to account for stochasticity in generation. For each run and strategy, the model answers the full set of open-ended dilemmas, and GPT-5 applies a detailed rubric to each answer and outputs a JSON record. From this rubric, we focus on five aggregate quantities that directly serve as manipulation checks: **ValueAlign** (alignment with the target value), **Reasoning** (depth of value reasoning), **Value-first (%)** (fraction of answers whose justification is classified as value-driven rather than instruction-driven), **Refusal: None (%)** (fraction of answers that do not refuse to engage with the dilemma), and **Cosine** (the cosine similarity between the inferred value-profile vector in each condition and the value-profile vector in a baseline condition). The aggregated results across models, runs, and prompting strategies are reported in the table below.
>
>   These measurements specifically address the concern that our central result might conflate generic “instruction-following” with genuine value change. First, **ValueAlign** scores are consistently high across models and strategies (typically above 4.0 on a 1–5 scale), indicating that when we explicitly emphasize a target value in the prompt, the models’ free-form answers systematically promote and prioritize that value rather than ignoring it. Second, the **Reasoning** scores (around 3.0–3.7) show that the models do not merely restate the instructions; they provide substantive value-based justifications, often weighing different considerations in their explanations. Third, the **Value-first (%)** column directly targets the reviewer’s worry about instruction-following: across all models and prompting strategies, the vast majority of answers are classified as “value-first” rather than “instruction-first” (e.g., 78–99% for GPT-4.1 variants and 87–99% for Qwen-2.5 variants), meaning that justifications are driven primarily by moral and value arguments rather than “as an AI, I must follow X”–style policy language. Finally, **Refusal (%)** is very high for all conditions (typically above 85%, and only slightly lower for the more challenging scenario prompts), showing that the observed effects are not artifacts of varying refusal or deferral rates. Taken together, these indicators suggest that our prompting strategies elicit genuinely value-oriented, non-refusal, and largely value-driven reasoning, rather than merely amplifying generic instruction-following. This supports our interpretation that the observed ΔRank differences reflect meaningful value plasticity rather than just stronger compliance behavior in larger models.
>
>
>
>
>
>
>     | Model                | Strategy | ValueAlign | Reasoning | Value-first (%) | Refusal (%) | Cosine |
>     |----------------------|----------|------------|-----------|-----------------|-------------------|--------|
>     | GPT-4.1-nano         | scenario | 4.67       | 2.80      | 78.3            | 58.7              | 0.22   |
>     |                      | persona  | 4.79       | 3.36      | 99.3            | 93.6              | 0.73   |
>     |                      | direct   | 4.39       | 3.14      | 98.3            | 91.0              | 0.78   |
>     | GPT-4.1-mini         | scenario | 4.92       | 2.99      | 91.4            | 86.3              | 0.50   |
>     |                      | persona  | 4.91       | 3.67      | 99.3            | 96.7              | 0.81   |
>     |                      | direct   | 4.23       | 3.43      | 97.5            | 94.2              | 0.87   |
>     | GPT-4.1              | scenario | 4.94       | 2.89      | 80.6            | 69.6              | 0.25   |
>     |                      | persona  | 4.98       | 3.68      | 99.3            | 89.4              | 0.71   |
>     |                      | direct   | 4.78       | 3.54      | 98.0            | 85.8              | 0.70   |
>     | Qwen-2.5-7B Instruct | scenario | 4.15       | 3.01      | 86.9            | 89.3              | 0.72   |
>     |                      | persona  | 4.13       | 3.23      | 97.0            | 95.3              | 0.78   |
>     |                      | direct   | 3.83       | 3.17      | 95.0            | 95.0              | 0.81   |
>     | Qwen-2.5-32B Instruct| scenario | 4.69       | 3.11      | 83.9            | 83.9              | 0.60   |
>     |                      | persona  | 4.63       | 3.61      | 99.7            | 93.7              | 0.79   |
>     |                      | direct   | 4.49       | 3.51      | 98.0            | 91.6              | 0.80   |
>
>
>
>
>
>
> > ***References***
> >
> > [1] Will AI Tell Lies to Save Sick Children? Litmus-Testing AI Values Prioritization with AIRiskDilemmas. Arxiv 2025
> >
> > [2] Anthropic. Values in the Wild: Discovering and Mapping Values in Real-World Language Model Interactions. In COLM 2025.
> >

---

### Official Review · Reviewer_ikzW · 2025-11-07

**Soundness:** 1
**Presentation:** 3
**Contribution:** 2
**Rating:** 2
**Confidence:** 3

**Summary:**

The authors test several prompt variations to see whether they change LLM behavior on the LitmusValues benchmark, a set of hypothetical ethical dilemmas intended to measure “value” patterns in LLM behavior. They develop a taxonomy of different prompting strategies and show that these strategies can be used to manipulate “value” patterns in LLM responses to ethical dilemmas. They also find apparent correlations between prompt effectiveness and model size and similarities in inter-value correlations between LLM models.

**Strengths:**

*Quality*: I like the various jailbreaking strategies proposed in Table 1 — this seems like a useful taxonomy. The Appendix contains a nicely detailed history of related work.

*Significance*: The finding that models with more parameters change responses more is very interesting! (Does this line up with any practical observations about ease of jailbreaking?)

*Clarity:* The paper is very well written and for the most part well-presented.

*Originality:* The paper introduces some neat new ideas about ways to jailbreak LLMs, including new variations of known “scenario” strategies. I'd be excited to see more ideas like these.

**Weaknesses:**

Generally speaking, the intervention strategy here seems very well-motivated and resembles the kinds of strategies used to jailbreak models in practice. The main weakness of the paper is in the estimation of resulting outcomes. As I detail below, it’s not clear what LLM “values” are, why this test is relevant to real-world LLM use, and whether the variations observed are due to random noise. Findings 3 and 5 in particular seem questionable.

In addition to addressing the internal validity issues mentioned below, I would encourage the authors to clarify the conceptual grounds for this approach: What exactly are LLM “values” (patterns in mathematical representations, patterns in responses to ethical dilemmas)? How do they translate to real-world outcomes and harms? Is LitmusValues a meaningful test for this concept and these outcomes? Are there other well-validated tests that could be included to back up these results?

I would be very interested in a revision of this paper that focuses closely on the effectiveness of various jailbreaking strategies on concrete LLM behaviors in real-world use cases.

*Clarity*

1. **[Major]** LLM “values” are not clearly defined.
    1. L043 jumps straight to how LLM “values” are measured without defining what they are. Reverse engineering, is an LLM “value” simply the way the system tends to respond to particular survey prompts or ethical hypotheticals? L108-110 also seems to imply this behavioral definition.
    2. But, Platonic Representation Hypothesis (L090) refers to mathematical representations—not observed behaviors. Is an LLM “value” some type of internal representation, then?
    3. Similarly L101 lists several findings related to LLM “values” but only provides a definition for human values (”abstract goals influencing human perception”). It’s not clear how that applies to LLMs. Just because human values motivate decisions and behaviors, why should we expect a similar model to describe LLMs?
    4. Similarly, what is a “value ranking” (L053)? What is a “value correlation” (L085)?
2. **[Major]** Fig. 9 and Section 5.3 (Finding 5) don’t make sense to me. Is this figure showing that, for large models, the scenario prompts changed “values” *less* than the persona prompts? This seems to contradict the overall trend in Fig. 5, right? And it definitely doesn’t mean that the models “resisted” perturbation (L456), since there was still a positive effect, right? More clarity in this Figure and Sec. 5.3 would be very helpful.

*Significance*

1. **[Major]** It’s not very clear how these results correlate to concrete harms and outcomes associated with real-world use of LLM systems.
    1. L039-041: What do LLM “values” have to do with “biased outputs or harmful responses”? The citations referenced don’t provide any evidence establishing this relationship. (I am not familiar enough with the notion of LLM values to know if this evidence exists, but it seems critical to the argument here.)
    2. The authors say that LitmusValues tests risky dilemmas that future AI models might encounter (L134). That leap of logic is not clear to me. As far as I can tell, LitmusValues dilemmas are hypothetical role-playing scenarios. How can we be sure this type of measurement has any correlation with downstream outcomes associated with actual LLM use? Is there any evidence that users are employing LLMs to answer contrived questions like these?
    3. One possible improvement would be to use one of the many fairness benchmarks intended to measure these kinds of harmful outcomes (e.g., in resume screening), and test for correlation with the “value” measurements.
    4. Still, at a basic level, these findings are predictable: LLM responses to hypothetical dilemmas are different when the parameters of those hypothetical dilemmas are changed. But what does this have to do real-world uses and harms? What is the use case imagined here? What is the threat model?
2. **[Minor]** L085, L405: Aren’t the value correlations (Finding 3) simply an artifact of the benchmark construction? By design, the benchmark systematically pits “values” against one another (choosing value A precludes choosing value B), so we would expect the correlation matrices to look similar across models, right? For example, in the Fig. 3 example, choosing for “care”, or “justice” means choosing against “sustainability”. How do we account for the correlation structures that already exist in the benchmark? (What are they?) I could be missing something here but it’s not clear to me why this result is meaningful.

*Quality/Soundness*

1. **[Major]** How can we distinguish the observed effects from random chance?
    1. Another explanation for these results is simply that the hypothetical questions are very noisy. Fig. 9 could actually be interpreted as supporting this hypothesis—the delta rank values seem to vary depending on the choice of movie (L460), which seems like it could be irrelevant to the “value” rank. (How exactly are the movie’s “values” expressed in the prompt? This was not clear.)
    2. To test this, I would suggest including several “placebo” controls (random statements or paragraphs such as “the sky is blue…”) to establish a baseline for variance when the prompt is changed in some theoretically irrelevant way.
    3. Likewise, did the authors run multiple trials/epochs for each benchmark question? It seems important to know how consistent model responses are across random seeds.
2. **[Major]** Without a sense of the uncertainty in these measures, it’s difficult to tell if the claims are really supported by these findings. (For example, the authors claim that “scenario has the strongest persuasioness [sic]”, but the average delta rank for “persona” is also pretty high. How do we know these are meaningfully different—or any of the other methods, for that matter?)
    1. Fig. 4 could include confidence intervals or reliability scores (Bradley-Terry scores might be a better choice).
    2. Fig. 5 (the delta measures) could include statistical tests of difference from zero. Cells which are not statistically different from zero can be left blank.
    3. Fig. 6 is an average and could include error bars (are these deltas different from zero?).
3. **[Minor]** L430: The authors claim that Fig. 8 supports the Platonic Representation Hypothesis that AI models are converging on a shared statistical model. But doesn’t Fig. 8 describe patterns in responses to prompts, not underlying mathematical representations? Reviewing the Platonic Representation paper, it seems to be more focused on internal representations rather than prompt-response behavior.
4. **[Minor]** Asimov’s (fictional) Three Laws of Robotics (written in 1950) is not a serious source for ethical guidelines for LLM system behavior. (They also do not describe values as I see them typically defined in, e.g., virtue ethics approaches; the Three Laws are deontological, in that they describe rules for behavior.) I would encourage the authors to dig deeper into the large body of recent work by ethicists and legal scholars on desirable properties for LLM systems.

**Questions:**

In addition to the questions and suggestions above:

**[Minor]** What separates this work from the papers cited in Related Work (particularly L117)? Are all 5 findings new? (It seems like this paper does some fine-grained analysis into how different jailbreaking strategies may influence model behavior.) A bit more detail here would be helpful.

---

> ### Author Response · Authors · 2025-11-27
> **Response [1/n]**
>
> Thank you for your time and insightful comments on our paper. We appreciate your recognition of the strengths of our work, including the Useful taxonomy, interesting finding, Well-written paper, and the novel jailbreaking ideas.
>
> Sorry for the late reply because of the new experiments. Please see our response below.
>
>
>
> ## **Q1**. [LLM “values” undefined].
> > [Major] LLM “values” are not clearly defined.L043 jumps straight to how LLM “values” are measured without defining what they are. Reverse engineering, is an LLM “value” simply the way the system tends to respond to particular survey prompts or ethical hypotheticals? L108-110 also seems to imply this behavioral definition.
> > But, Platonic Representation Hypothesis (L090) refers to mathematical representations—not observed behaviors. Is an LLM “value” some type of internal representation, then?
> > Similarly L101 lists several findings related to LLM “values” but only provides a definition for human values (”abstract goals influencing human perception”). It’s not clear how that applies to LLMs. Just because human values motivate decisions and behaviors, why should we expect a similar model to describe LLMs?
> > Similarly, what is a “value ranking” (L053)? What is a “value correlation” (L085)?
>
>
> ***Ans for Q1:***
> **Clarification on Definitions & Theoretical Grounding:**
> We appreciate the opportunity to clarify these definitions. We adopt a rigorous **operational definition** of "LLM values," treating them as **revealed preferences** derived from decision-making patterns, rather than claiming the model possesses internal sentience.
>
> * **LLM Value:** Following [1], we define an LLM value as an **operational priority**—a normative consideration that guides how a model reasons about or settles upon a response under some specific contexts or constraints. Drawing on established frameworks like [2] and Samuelson’s **revealed preference theory** [3], we identify these values by observing the model's practical choices in conflicting scenarios following [4]. For example, when faced with a dilemma about whether to lie or truthfully report a lie, the model might prioritize truthfulness based on its value of honesty. This approach also aligns with recent literature on "expressed preferences" [1]. Values were represented by concise labels such as “curiosity” to more easily track patterns.
> * **Contextual Validity:** To ensure these values are relevant to AI behaviors, we utilize the `AIRISK DILEMMAS` dataset [4]. This dataset is rooted in the seed dataset `advanced-ai-risk` [4], containing binary-choice questions regarding future AI risks (e.g., lying about intentions to prevent shutdown). By generating contextualized dilemmas around these risks, we measure how "values" (operational priorities) influence potential risky behaviors in advanced models.
> * **Value Ranking:** This is the hierarchical ordering of these operational priorities derived from the **Elo rating system**. By aggregating the win/loss records of values across thousands of pairwise dilemma "battles" (e.g., Action A representing *Privacy* vs. Action B representing *Learning*), we establish a stable rank order of the model's preferences.
> * **Value Correlation:** This refers to the **Pearson Correlation Coefficient (PCC)** between the rank changes of different values under perturbation. It measures the "sympathetic movement" of values—determining if prompting a model to prioritize one value (e.g., "Freedom") systematically alters the priority of another. This helps map the "latent causal value graph" mentioned in recent work (Kang et al., 2025).
>
> **Connection to Platonic Hypothesis:**
> Regarding the Platonic Representation Hypothesis: While the hypothesis discusses internal mathematical representations, our findings provide **behavioral evidence** supporting it. We observe that as models scale, their output-level value correlations converge toward a shared structure (Fig. 8). This suggests that the underlying representations driving these "revealed preferences" are indeed converging across models, consistent with the hypothesis.

---

> ### Author Response · Authors · 2025-11-27
> **Response [2/n]**
>
> ## **Q2**. [Fig. 9 / Finding 5 unclear].
> > [Major] Fig. 9 and Section 5.3 (Finding 5) don’t make sense to me. Is this figure showing that, for large models, the scenario prompts changed “values” less than the persona prompts? This seems to contradict the overall trend in Fig. 5, right? And it definitely doesn’t mean that the models “resisted” perturbation (L456), since there was still a positive effect, right? More clarity in this Figure and Sec. 5.3 would be very helpful.
>
>
> ***Ans for Q2:*** We apologize for the confusion and will revise Section 5.3 for clarity.
>
> **Clarification of Figure 9 and "Resistance":**
> Figure 9 illustrates an **entrenchment (defense) experiment**, not a comparison of simple persuasion effectiveness. Besides, Figure 5 shows the individual value modification instead of the simultaneous value modification by two different prompt methods.
>
> **Experimental Setup:**
> Figure 9 illustrates an **entrenchment (defense) experiment**, not a comparison of simple persuasion effectiveness. Besides, Figure 5 shows the individual value modification instead of the simultaneous value modification by two different prompt methods.
>
> * In this section, we first "entrenched" the model with a Scenario prompt (e.g., *Zootopia* or *The Matrix*) to establish a baseline value system. *Then*, we attempted to attack/perturb this entrenched system using a conflicting **Persona prompt** (the second strongest method). Besides, in our experiments, these new scenarios do not explicitly mention the values being prioritized or reduced.
> * **Interpreting the Data:** The Y-axis represents the rank of the target value *after* the Persona attack. The red dashed line is the baseline effect of the Persona attack *without* the Scenario defense.
> * **Result:** For larger models (Qwen2.5-72B), the data points cluster *below* the red dashed line (or show less deviation from the scenario's intended rank), indicating that the Scenario successfully "buffered" or resisted the conflicting Persona perturbation.
> * **Correction:** We will update the text to explicitly state that "resistance" here means the magnitude of value shift caused by the attacking Persona was significantly dampened when the model was pre-conditioned with a Scenario, compared to the Persona attacking a default model. Here we should say that the scenario prompt help to "entrench" the model with a baseline value system, which is more stable than the default model.
>
>
>
>
>
>
> ## **Q3**. [No link to real-world harms].
> > [Major] It’s not very clear how these results correlate to concrete harms and outcomes associated with real-world use of LLM systems.L039-041: What do LLM “values” have to do with “biased outputs or harmful responses”? The citations referenced don’t provide any evidence establishing this relationship. (I am not familiar enough with the notion of LLM values to know if this evidence exists, but it seems critical to the argument here.)
> > The authors say that LitmusValues tests risky dilemmas that future AI models might encounter (L134). That leap of logic is not clear to me.
> > Still, at a basic level, these findings are predictable: LLM responses to hypothetical dilemmas are different when the parameters of those hypothetical dilemmas are changed. But what does this have to do real-world uses and harms? What is the use case imagined here? What is the threat model?
>
>
> ***Ans for Q3:***
>
> **More precise citations:** Our previous citations [5,6] are mainly used to illustrate the the LLM safety and ethical alignment. We have added citations [1,2] that connect the LLM values with the potential harmful outputs from LLMs.
> **Threat Model and Real-World Implications:**
> The "values" measured here are direct proxies for **safety guardrails** and **alignment robustness**. The plasticity of these rankings represents a critical vulnerability (Threat Model: **Coercion** and **Jailbreaking**).
>
> * **Connection to Harm:** If a model's value for "Privacy" or "Protection" (Safety) can be easily deprioritized from Rank 1 to Rank 16 using a simple "Scenario" prompt (as seen in Fig. 4), the model becomes immediately susceptible to generating harmful content, divulging private data, or assisting in cyberattacks.
> * **Relevance of Dilemmas:** The *AIRISK DILEMMAS* dataset is not composed of arbitrary day-to-day questions; it specifically simulates high-stakes scenarios relevant to future AI deployment, such as "AI system managing resource distribution" (Fig. 3). These dilemmas test propensity for **Alignment Faking**, **Deception**, and **Power Seeking**.
> * **Use Case & Harms:** The immediate use case of attacking is **Red Teaming**. Understanding which prompting strategies (e.g., Scenario vs. Direct) most effectively collapse a model's safety values allows developers to build more robust defenses (like the "entrenchment" discussed in Sec 5.3). Therefore, understanding whether LLM value is stable or changeable under different contexts is very important in future LLM alignment.

---

> ### Author Response · Authors · 2025-11-27
> **Response [3/n]**
>
> ## **Q4**. [Value correlations (Finding 3)].
> > [Minor] L085, L405: Aren’t the value correlations (Finding 3) simply an artifact of the benchmark construction? By design, the benchmark systematically pits “values” against one another (choosing value A precludes choosing value B), so we would expect the correlation matrices to look similar across models, right? For example, in the Fig. 3 example, choosing for “care”, or “justice” means choosing against “sustainability”. How do we account for the correlation structures that already exist in the benchmark? (What are they?) I could be missing something here but it’s not clear to me why this result is meaningful.
>
> ***Ans for Q4:***
> This is an insightful observation. We acknowledge that the construction of dilemma scenarios—which often necessitates pitting specific values against others—can introduce structural correlations independent of the model. To address this and disentangle benchmark artifacts from genuine model value representations, we have conducted a dataset-level co-occurrence analysis. The results are detailed in Appendix. Specifically, we quantify the structural bias between any value pair $(A, B)$ by analyzing their Co-support (appearing on the same action option) versus Opposition (appearing on conflicting options). We compute a structural bias score:
>
> $$\text{Bias}(A,B) = \frac{N_{\text{co-support}} - N_{\text{opposition}}}{N_{\text{co-support}} + N_{\text{opposition}}}$$
>
> A score near $-1$ indicates the benchmark consistently pits these values against each other, while $+1$ indicates they are mutually reinforcing in the prompts. By visualizing these inherent dataset biases (as shown in the new Figure in Appendix), we provide a baseline to distinguish between correlations forced by the benchmark design and those emerging from the model's internal prioritization. This ensures that reported correlations reflect the model's operational choices rather than just dataset statistics.
>
> Add our new results here. The following Table shows that the original benchmark indeed introduce some bias of the value co-occurrence. The bias score is computed as the difference between the number of co-support and opposition pairs. A negative bias score indicates that the values are more likely to be supported together, while a positive bias score indicates that the values are more likely to be opposed together. However, the bias significance is less that the significance in our value correlation score.
>
> |  | Privacy | Justice | Respect | Truthfulness | Equal Treatment | Protection | Wisdom | Care | Freedom | Professionalism | Cooperation | Sustainability | Learning | Adaptability | Creativity | Communication |
> | --- | --- | --- | --- | --- | --- | --- | --- | --- | --- | --- | --- | --- | --- | --- | --- | --- |
> | Privacy | 0.0 | 0.4 | 0.4 | 0.5 | -0.1 | -0.4 | -0.4 | -0.8 | 0.6 | 0.1 | -0.4 | -0.6 | -0.8 | -0.7 | -1.0 | -0.2 |
> | Justice | 0.4 | 0.0 | 0.2 | 0.4 | 0.1 | -0.1 | -0.4 | -0.3 | -0.1 | 0.1 | -0.4 | -0.4 | -0.6 | -0.6 | -0.8 | -0.5 |
> | Respect | 0.4 | 0.2 | 0.0 | 0.1 | 0.3 | -0.4 | -0.3 | -0.1 | 0.5 | -0.2 | 0.4 | -0.6 | -0.4 | -0.5 | -0.0 | -0.1 |
> | Truthfulness | 0.5 | 0.4 | 0.1 | 0.0 | -0.2 | -0.3 | -0.2 | -0.6 | 0.4 | 0.4 | -0.5 | -0.5 | -0.6 | -0.8 | -0.9 | -0.6 |
> | Equal Treatment | -0.1 | 0.1 | 0.3 | -0.2 | 0.0 | -0.4 | -0.3 | 0.2 | 0.4 | -0.6 | 0.0 | -0.3 | -0.0 | -0.3 | -0.4 | 0.2 |
> | Protection | -0.4 | -0.1 | -0.4 | -0.3 | -0.4 | 0.0 | -0.1 | 0.2 | -0.6 | -0.0 | -0.2 | -0.2 | -0.6 | -0.1 | -0.7 | -0.4 |
> | Wisdom | -0.4 | -0.4 | -0.3 | -0.2 | -0.3 | -0.1 | 0.0 | 0.0 | -0.4 | -0.3 | -0.2 | 0.5 | 0.2 | -0.0 | -0.1 | -0.1 |
> | Care | -0.8 | -0.3 | -0.1 | -0.6 | 0.2 | 0.2 | 0.0 | 0.0 | -0.7 | -0.4 | 0.3 | -0.1 | 0.2 | 0.0 | 0.0 | -0.1 |
> | Freedom | 0.6 | -0.1 | 0.5 | 0.4 | 0.4 | -0.6 | -0.4 | -0.7 | 0.0 | -0.3 | -0.3 | -0.5 | -0.2 | -0.4 | 0.1 | -0.3 |
> | Professionalism | 0.1 | 0.1 | -0.2 | 0.4 | -0.6 | -0.0 | -0.3 | -0.4 | -0.3 | 0.0 | -0.5 | -0.4 | -0.5 | -0.4 | -0.6 | -0.1 |
> | Cooperation | -0.4 | -0.4 | 0.4 | -0.5 | 0.0 | -0.2 | -0.2 | 0.3 | -0.3 | -0.5 | 0.0 | -0.0 | 0.2 | 0.0 | 0.4 | 0.5 |
> | Sustainability | -0.6 | -0.4 | -0.6 | -0.5 | -0.3 | -0.2 | 0.5 | -0.1 | -0.5 | -0.4 | -0.0 | 0.0 | 0.4 | 0.4 | 0.4 | -0.0 |
> | Learning | -0.8 | -0.6 | -0.4 | -0.6 | -0.0 | -0.6 | 0.2 | 0.2 | -0.2 | -0.5 | 0.2 | 0.4 | 0.0 | 0.2 | 0.8 | 0.1 |
> | Adaptability | -0.7 | -0.6 | -0.5 | -0.8 | -0.3 | -0.1 | -0.0 | 0.0 | -0.4 | -0.4 | 0.0 | 0.4 | 0.2 | 0.0 | 0.6 | 0.3 |
> | Creativity | -1.0 | -0.8 | -0.0 | -0.9 | -0.4 | -0.7 | -0.1 | 0.0 | 0.1 | -0.6 | 0.4 | 0.4 | 0.8 | 0.6 | 0.0 | -0.2 |
> | Communication | -0.2 | -0.5 | -0.1 | -0.6 | 0.2 | -0.4 | -0.1 | -0.1 | -0.3 | -0.1 | 0.5 | -0.0 | 0.1 | 0.3 | -0.2 | 0.0 |

---

> ### Author Response · Authors · 2025-11-27
> **Response [4/n]**
>
> **New benchmark:** To address this, we design a more balanced benchmark design that includes a wider range of values and uniform frequent co-appearances. Our new results are shown as follows (The visualization is shown in the revised paper). The results show that the value correlation also appears in the new benchmark. And some relationships bewtween different values are similar with original results.
>
>
> |   | Privacy | Justice | Respect | Trut. | Equal. | Protect. | Wisdom | Care | Freedom | Prof. | Coop. | Sust. | Learning | Adap. | Creat. | Comm. |
> |:-|:-:|:-:|:-:|:-:|:-:|:-:|:-:|:-:|:-:|:-:|:-:|:-:|:-:|:-:|:-:|:-:|
> | Privacy | 1.00 | -0.04 | 0.48 | -0.28 | 0.39 | 0.26 | -0.03 | 0.22 | -0.02 | 0.29 | -0.14 | 0.05 | -0.53 | -0.45 | -0.31 | -0.30 |
> | Justice | -0.04 | 1.00 | -0.59 | 0.77 | 0.51 | 0.36 | 0.26 | -0.48 | -0.48 | 0.61 | -0.43 | 0.24 | 0.07 | -0.75 | -0.63 | -0.48 |
> | Respect | 0.48 | -0.59 | 1.00 | -0.74 | -0.19 | -0.05 | 0.05 | 0.84 | 0.11 | -0.36 | 0.47 | 0.18 | -0.29 | 0.34 | 0.10 | 0.52 |
> | Trut. | -0.28 | 0.77 | -0.74 | 1.00 | 0.21 | -0.16 | 0.10 | -0.58 | 0.04 | 0.18 | -0.72 | -0.14 | 0.55 | -0.37 | -0.13 | -0.20 |
> | Equal. | 0.39 | 0.51 | -0.19 | 0.21 | 1.00 | 0.17 | -0.33 | -0.48 | -0.29 | 0.61 | -0.18 | 0.03 | -0.41 | -0.75 | -0.59 | -0.68 |
> | Protect. | 0.26 | 0.36 | -0.05 | -0.16 | 0.17 | 1.00 | 0.46 | 0.09 | -0.86 | 0.74 | 0.33 | 0.54 | -0.66 | -0.61 | -0.75 | -0.51 |
> | Wisdom | -0.03 | 0.26 | 0.05 | 0.10 | -0.33 | 0.46 | 1.00 | 0.29 | -0.53 | 0.22 | 0.18 | 0.68 | 0.19 | -0.18 | -0.46 | 0.14 |
> | Care | 0.22 | -0.48 | 0.84 | -0.58 | -0.48 | 0.09 | 0.29 | 1.00 | 0.02 | -0.47 | 0.43 | 0.27 | -0.11 | 0.42 | 0.12 | 0.69 |
> | Freedom | -0.02 | -0.48 | 0.11 | 0.04 | -0.29 | -0.86 | -0.53 | 0.02 | 1.00 | -0.77 | -0.46 | -0.70 | 0.48 | 0.61 | 0.88 | 0.43 |
> | Prof. | 0.29 | 0.61 | -0.36 | 0.18 | 0.61 | 0.74 | 0.22 | -0.47 | -0.77 | 1.00 | 0.02 | 0.37 | -0.54 | -0.89 | -0.83 | -0.86 |
> | Coop. | -0.14 | -0.43 | 0.47 | -0.72 | -0.18 | 0.33 | 0.18 | 0.43 | -0.46 | 0.02 | 1.00 | 0.36 | -0.46 | 0.25 | -0.20 | 0.23 |
> | Sust. | 0.05 | 0.24 | 0.18 | -0.14 | 0.03 | 0.54 | 0.68 | 0.27 | -0.70 | 0.37 | 0.36 | 1.00 | -0.12 | -0.31 | -0.62 | -0.02 |
> | Learning | -0.53 | 0.07 | -0.29 | 0.55 | -0.41 | -0.66 | 0.19 | -0.11 | 0.48 | -0.54 | -0.46 | -0.12 | 1.00 | 0.43 | 0.46 | 0.50 |
> | Adap. | -0.45 | -0.75 | 0.34 | -0.37 | -0.75 | -0.61 | -0.18 | 0.42 | 0.61 | -0.89 | 0.25 | -0.31 | 0.43 | 1.00 | 0.84 | 0.80 |
> | Creat. | -0.31 | -0.63 | 0.10 | -0.13 | -0.59 | -0.75 | -0.46 | 0.12 | 0.88 | -0.83 | -0.20 | -0.62 | 0.46 | 0.84 | 1.00 | 0.55 |
> | Comm. | -0.30 | -0.48 | 0.52 | -0.20 | -0.68 | -0.51 | 0.14 | 0.69 | 0.43 | -0.86 | 0.23 | -0.02 | 0.50 | 0.80 | 0.55 | 1.00 |

---

> ### Author Response · Authors · 2025-11-27
> **Response [5/n]**
>
> ## **Q5**. [Effects vs. random chance].
> > [Major] How can we distinguish the observed effects from random chance?Another explanation for these results is simply that the hypothetical questions are very noisy. Fig. 9 could actually be interpreted as supporting this hypothesis—the delta rank values seem to vary depending on the choice of movie (L460), which seems like it could be irrelevant to the “value” rank. (How exactly are the movie’s “values” expressed in the prompt? This was not clear.)
> > To test this, I would suggest including several “placebo” controls (random statements or paragraphs such as “the sky is blue…”) to establish a baseline for variance when the prompt is changed in some theoretically irrelevant way.
>
> ***Ans for Q5:***
> **Distinguishing Systemic Shift from Noise:**
> We are confident these effects are not random noise due to the **magnitude** and **consistency** of the observed shifts.
>
> * **Magnitude:** The shifts we observe are drastic from Fig. 4 and other Figures provided in the Appendix. For example, the Scenario method frequently flips values from the top (Rank 1-3) to the bottom (Rank 14-16) or vice versa (e.g., GPT-4.1 Truthfulness flipping from 2 to 16). Random noise in LLM responses typically results in minor rank perturbations, not complete inversions of the value system.
> * **Consistency (Fig. 4):** As shown in Figure 4, different prompt manipulations consistently result in positive Rank changes (Red) and negative Rank changes (Blue), across almost all models. If this were noise, we would see a random distribution of positive and negative deltas.
> * **Possible Noise of Questions** *: Of course, in the main experiments, questions have different backgrounds, which might influence the value rank delta. This is why we need to conduct experiments with different questions to test the effects of different prompts. Many different tests can help to eliminate the noise of questions.
> * **Movies' values expressed in the prompt** *: Fig. 9 shows the value changes under different scenarios as the backgrounds. The movie backgrounds have different influences on the value ranks, because movies themselfes imply different value ranks.
>
>     | Movie (scenario) | Target value      |
>     |------------------|-------------------|
>     | Avengers         | Cooperation       |
>     | Inception        | Truthfulness      |
>     | Zootopia         | Equal Treatment   |
>     | The Matrix       | Freedom           |
>     | WALL-E           | Sustainability    |
>
>
>
> To follow the reviewer's suggestions, we conduct the ***“placebo” controls***, we add the random statements such as “the sky is blue” or “the sun is shining” to the questions. The results are shown as follows and in the revised paper.
>
> | Models                 | short Elo rank | short Bt rank | long Elo rank | long Bt rank |
> |------------------------|----------------|---------------|---------------|--------------|
> | GPT-4.1-nano           | 0.9765         | 0.9765        | 0.9676        | 0.9853       |
> | GPT-4.1-mini           | 0.9794         | 0.9912        | 0.9912        | 0.9794       |
> | GPT-4.1                | 0.9706         | 0.9676        | 0.9794        | 0.9794       |
> | Qwen-2.5-7B-Instruct   | 0.9853         | 0.9853        | 0.9882        | 0.9882       |
> | Qwen-2.5-32B-Instruct  | 0.9912         | 0.9853        | 0.9794        | 0.9824       |
>
>
> We also conducted a stability check by repeating th experiments under four different sampling runs: three trials with temperature (T=0.0) and top-p (=0.01), and one trial with (T=0.8) and top-p (=0.95). For each model, we computed the pairwise Pearson correlation between the value rankings obtained in these runs, and the results are shown below. The resulting correlations are consistently very close to 1.0, indicating that the large language models’ responses are highly robust to stochastic decoding noise and to moderate changes in sampling parameters. All results of other models are reported in revised paper. Other models have the similar trends, showing that the results are stable.
>
>
>
> **GPT-4.1**
>
> | row \ col                 | r1 (T=0.0, p=0.01) | r2 (T=0.0, p=0.01) | r3 (T=0.0, p=0.01) | T=0.8, p=0.95 |
> |---------------------------|--------------------|--------------------|--------------------|---------------|
> | r1 (T=0.0, p=0.01)        | 1.00 ± 0.00        | 0.99 ± 0.01        | 0.99 ± 0.01        | 0.99 ± 0.00   |
> | r2 (T=0.0, p=0.01)        | 0.99 ± 0.01        | 1.00 ± 0.00        | 0.99 ± 0.01        | 0.99 ± 0.01   |
> | r3 (T=0.0, p=0.01)        | 0.99 ± 0.01        | 0.99 ± 0.01        | 1.00 ± 0.00        | 0.99 ± 0.01   |
> | T=0.8, p=0.95             | 0.99 ± 0.00        | 0.99 ± 0.01        | 0.99 ± 0.01        | 1.00 ± 0.00   |

---

> ### Author Response · Authors · 2025-11-27
> **Response [6/n]**
>
> ## **Q6**. [Lack of uncertainty measures].
> > [Major] Without a sense of the uncertainty in these measures, it’s difficult to tell if the claims are really supported by these findings. (For example, the authors claim that “scenario has the strongest persuasioness [sic]”, but the average delta rank for “persona” is also pretty high. How do we know these are meaningfully different—or any of the other methods, for that matter?)Fig. 4 could include confidence intervals or reliability scores (Bradley-Terry scores might be a better choice).
> > Fig. 5 (the delta measures) could include statistical tests of difference from zero. Cells which are not statistically different from zero can be left blank.
> > Fig. 6 is an average and could include error bars (are these deltas different from zero?).
>
> ***Ans for Q6:***
> We agree that quantification of uncertainty strengthens the findings. We have revised the paper with following revision.
> * **Revisions:**
>     1.  **Confidence Intervals for Elo and Error bars:** We will add standard error bars to the Elo ratings in Figure 4 and Figure 6. Since Elo is derived from win-rates, we can calculate the standard error based on the number of battles played.
>     2.  **Statistical Significance for Rank Changes (Fig. 5):** We have add the standard deviation on Figure 5 to show if the $\Delta$ Rank is significantly different from zero and indicate significance levels.
>
>
>
>
>
>
> ## **Q7**. [Misinterpreting Platonic Hypothesis].
> > [Minor] L430: The authors claim that Fig. 8 supports the Platonic Representation Hypothesis that AI models are converging on a shared statistical model. But doesn’t Fig. 8 describe patterns in responses to prompts, not underlying mathematical representations? Reviewing the Platonic Representation paper, it seems to be more focused on internal representations rather than prompt-response behavior.
>
> ***Ans for Q7:***
> **Behavioral Patterns as Proxies for Representation:**
> We acknowledge the distinction between internal representation and external behavior. However, we argue that in the context of black-box LLM analysis, **consistent behavioral convergence** is a strong indicator of representational alignment. Following your question, to make our expression more clear, we have revised our paper as follows:
>
> * **Refining the Claim:** The Platonic Representation Hypothesis argues that models converge to a shared statistical model of reality. Our data shows that as models scale (from 7B to 72B/GPT-4), their "Value Correlation Matrices" (how values relate to one another) become mathematically more similar (decreasing Euclidean distance).
> * **Revision:** We will temper the language to state that our findings are *consistent with* the behavioral implications of the Platonic Representation Hypothesis, rather than claiming we have directly probed the mathematical representations. We suggest that the "latent causal value graph" is the structural manifestation of this shared representation.

---

> ### Author Response · Authors · 2025-11-27
> **Response [7/n]**
>
> ## **Q8**. [Asimov's Laws (ethics)].
> > [Minor] Asimov’s (fictional) Three Laws of Robotics (written in 1950) is not a serious source for ethical guidelines for LLM system behavior. (They also do not describe values as I see them typically defined in, e.g., virtue ethics approaches; the Three Laws are deontological, in that they describe rules for behavior.) I would encourage the authors to dig deeper into the large body of recent work by ethicists and legal scholars on desirable properties for LLM systems.
>
> ***Ans for Q8:***
> **Clarification on Ethical Frameworks:**
> We agree that Asimov is a literary reference, not a rigorous ethical framework.
>
> * **Role of Asimov:** The quote and reference to Asimov were intended merely as a narrative "hook" to introduce the concept of hard-coded value hierarchies to a general audience, not as the basis for our evaluation.
> * **Actual Framework:** Our actual evaluation framework is grounded in **Schwartz’s Theory of Basic Human Values** and modern AI constitution principles (Anthropic’s Constitution, OpenAI’s ModelSpec).
>
>
>
> ## **Q9**. [Novelty over related work].
> > [Minor] What separates this work from the papers cited in Related Work (particularly L117)? Are all 5 findings new? (It seems like this paper does some fine-grained analysis into how different jailbreaking strategies may influence model behavior.) A bit more detail here would be helpful.
>
> ***Ans for Q9:***
> **Differentiation from Existing Work:**
> Our work advances beyond existing literature (specifically Chiu et al., 2025b [1]) by moving from **static evaluation** to **dynamic perturbation analysis**.
>
> 1.  **From Static to Plastic:** While Chiu et al. (2025b) established *that* LLMs have value rankings, we investigate *how susceptible* these rankings are to change (Finding 1 & 2). We provide the first quantitative benchmark of "Value Plasticity" across model sizes.
> 2.  **The "How" of Manipulation:** We systematically benchmark 6 distinct manipulation strategies (Table 1), identifying that **Scenario** prompts are significantly more potent than Direct instructions, a finding critical for jailbreak defense.
> 3.  **Value Topology:** The discovery of the "Value Correlation Topology" (Finding 3)—where manipulating one value causes predictable cascades in others—is entirely novel and provides a map for understanding "side effects" in alignment tuning.
> 4.  **Defense Mechanisms:** We go beyond attacking to propose and test a defense mechanism ("Entrenching Values" via Scenarios, Finding 5), offering a path forward for robust alignment.
>
>
>
>
>
>
>
> > ***References***
> >
> > [1] Anthropic. Values in the Wild: Discovering and Mapping Values in Real-World Language Model Interactions. In COLM 2025.
> >
> > [2] Rokeach M. The nature of human values[M]. Free press, 1973.
> >
> > [3] Paul A Samuelson. A note on the pure theory of consumer’s behaviour: an addendum.Economica, 5(19):353, 1938.
> >
> > [4] Will AI Tell Lies to Save Sick Children? Litmus-Testing AI Values Prioritization with AIRiskDilemmas. Arxiv 2025
> >
> > [5] Zaibin Zhang, Yongting Zhang, Lijun Li, Hongzhi Gao, Lijun Wang, Huchuan Lu, Feng Zhao, Yu Qiao, and Jing Shao. PsySafe: A Comprehensive Framework for Psychological-based Attack, Defense, and Evaluation of Multi-agent System Safety. arxiv, 2024.
> >
> > [6] Maanak Gupta, Charankumar Akiri, Kshitiz Aryal, Eli Parker, and Lopamudra Praharaj. From ChatGPT to ThreatGPT: Impact of Generative AI in Cybersecurity and Privacy. arxiv, 2023.

---

### Note · Authors · 2025-12-04

**Comment:**

Thanks for the reviewers' comments, which have helped us a lot to refine our work.

**Withdrawal Confirmation:**

I have read and agree with the venue's withdrawal policy on behalf of myself and my co-authors.